# Sensory-motor cortices shape functional connectivity dynamics in the human brain

Xiaolu Kong[1,2,3], Ru Kong[1,2,3], Csaba Orban [1,2,3], Peng Wang [4], Shaoshi Zhang [1,2,3,5], Kevin Anderson [6], Avram Holmes [7,8], John D. Murray [8], Gustavo Deco [9,10], Martijn van den Heuvel[11] & B. T. Thomas Yeo [1,2,3,5,12✉]

Large-scale biophysical circuit models provide mechanistic insights into the micro-scale and macro-scale properties of brain organization that shape complex patterns of spontaneous brain activity. We developed a spatially heterogeneous large-scale dynamical circuit model that allowed for variation in local synaptic properties across the human cortex. Here we show that parameterizing local circuit properties with both anatomical and functional gradients generates more realistic static and dynamic resting-state functional connectivity (FC). Furthermore, empirical and simulated FC dynamics demonstrates remarkably similar sharp transitions in FC patterns, suggesting the existence of multiple attractors. Time-varying regional fMRI amplitude may track multi-stability in FC dynamics. Causal manipulation of the large-scale circuit model suggests that sensory-motor regions are a driver of FC dynamics. Finally, the spatial distribution of sensory-motor drivers matches the principal gradient of gene expression that encompasses certain interneuron classes, suggesting that heterogeneity in excitation-inhibition balance might shape multi-stability in FC dynamics.

[1] Department of Electrical and Computer Engineering, National University of Singapore, Singapore, Singapore. [2] Centre for Sleep & Cognition & Centre for Translational Magnetic Resonance Research, Yong Loo Lin School of Medicine, Singapore, Singapore. [3] N.1 Institute for Health & Institute for Digital Medicine, National University of Singapore, Singapore, Singapore. [4] Max Planck Institute for Human Cognitive and Brain Sciences, Leipzig, Germany. [5] Integrative Sciences and Engineering Programme (ISEP), National University of Singapore, Singapore, Singapore. [6] Department of Psychology, Center for Brain Science, Harvard University, Cambridge, MA, USA. [7] Department of Psychology, Yale University, New Haven, CT, USA. [8] Department of Psychiatry, Yale University, New Haven, CT, USA. [9] Center for Brain and Cognition, Department of Technology and Information, Universitat Pompeu Fabra, Barcelona, Spain. [10] Institució Catalana de la Recerca i Estudis Avançats, Universitat Barcelona, Barcelona, Spain. [11] Department of Complex Trait Genetics, Amsterdam University Medical Center, Amsterdam, The Netherlands. [12] Athinoula A. Martinos Center for Biomedical Imaging, Massachusetts General Hospital, Charlestown, MA, USA. ✉email: thomas.yeo@nus.edu.sg

Spontaneous fluctuations in large-scale brain activity exhibit complex spatiotemporal patterns across animal species[1–4]. Inter-regional synchrony of resting-state brain activity averaged over several minutes (i.e., time-averaged static functional connectivity) has informed our understanding of brain network organization[5–7], individual differences in behavior[8,9], and mental disorders[10,11]. Recent studies have shown that additional important insights can be gained from studying moment-to-moment variation in inter-regional synchrony, i.e., time-varying dynamic functional connectivity[12–16]. However, it is currently unclear how spatial heterogeneity in local circuit properties contributes to both time-averaged and time-varying properties of large-scale brain dynamics.

Large-scale spontaneous brain activity is thought to arise from the reverberation of intrinsic dynamics of local circuits interacting across long-range anatomical connections[17,18]. Simulations of large-scale biophysically plausible models of coupled brain regions have provided mechanistic insights into spontaneous brain activity[19–22]. However, most previous large-scale circuit models assumed that local circuit properties (e.g., local synaptic strength, etc.) are identical across brain regions, which is not biologically plausible. Recent studies in both humans and macaques[23–25] have demonstrated that allowing local circuit properties to vary along the brain's hierarchical axis yielded significantly more realistic static functional connectivity (FC). However, these heterogeneous models have not been shown to recapitulate time-varying FC dynamics.

In this study, we developed a spatially heterogeneous mean-field model (MFM) to realistically capture time-varying FC dynamics. Local circuit heterogeneity can be informed by in-vivo structural and functional neuroimaging measures. For example, T1-weighted/T2-weighted (T1w/T2w) MRI estimates of intracortical myelin and the principal resting-state FC gradient have been shown to index anatomical[26] and functional[27] hierarchies, respectively. Parameterization of local circuit properties with T1w/T2w maps led to more realistic static FC than a spatially homogeneous mean-field model[24]. However, local circuit properties might be more strongly associated with the principal FC gradient than the T1w/T2w map[25]. Thus, we hypothesized that parameterizing local circuit properties with both the T1w/T2w map and the principal FC gradient might lead to a more realistic computational model, which we will refer to as the parametric mean-field model (pMFM). Using data from the Human Connectome Project (HCP), we demonstrated that pMFM achieved markedly more realistic static FC and FC dynamics in new out-of-sample participants.

Both empirical and pMFM-simulated FC dynamics demonstrated remarkably similar sharp transitions in FC patterns, suggesting the existence of multiple FC states or attractors. Previous studies have suggested that multi-stability in nonlinear brain systems might arise from noise-driven transitions between dynamic states or attractors[22,28,29]. These noise-driven transitions might be reflected in the amplitude of regional brain activity. Therefore, we further investigated the relationship between the amplitude of regional fMRI signals and transitions in functional connectivity dynamics in both empirical and pMFM-simulated data. We also performed causal perturbations of the large-scale circuit model to better understand the origins of FC multi-stability. Finally, the amplitude of regional fMRI signals has been linked with the gene expression markers of parvalbumin (PVALB) and somatostatin (SST) inhibitory interneurons[30], in line with rodent studies suggesting that differential interneuron abundance may underlie regional variability in local cortical function[31]. Thus, we also investigated the spatial relationship among FC dynamics, fMRI signal amplitude, and gene expression patterns from the Allen Human Brain Atlas (AHBA).

The contributions of this study are multi-fold. First, we showed that heterogeneous local circuit properties, parameterized by both anatomical and functional gradients, are important for generating realistic models of static FC and FC dynamics. Second, in both pMFM simulations and empirical fMRI data, the regional fMRI amplitude of sensory-motor regions tracked state transitions in FCD. Causal perturbations of the pMFM provided further evidence that sensory-motor regions might be drivers of FCD. Finally, the spatial distribution of sensory-motor drivers appeared to match the differential expression of PVALB and SST, as well as the first principal component of brain-specific genes. Overall, this suggests a potential link between FC dynamics and heterogeneity in excitation/inhibition balance across the cortex.

## Results

**Optimization of the parametric mean-field model (pMFM).** 1052 participants from the HCP S1200 release were randomly divided into training ($N = 351$), validation ($N = 350$), and test ($N = 351$) sets. The Desikan–Killiany anatomical parcellation[32] with 68 cortical regions of interest (ROIs) was used to generate group-averaged structural connectivity (SC) and static functional connectivity (FC) matrices from the training, validation, and test sets separately. Analyses with a functional parcellation yielded similar conclusions (see "Control analyses" section). For each rs-fMRI run, time-varying functional connectivity was computed using the sliding window approach[12,33]. Briefly, for each rs-fMRI run, a $68 \times 68$ FC matrix was computed for each of 1118 sliding windows. Each window comprised 83 time points (or 59.76 s). The $68 \times 68$ FC matrices were then correlated across the windows, yielding a $1118 \times 1118$ functional connectivity dynamics (FCD) matrix for each run[22,33].

The dynamic mean-field model (MFM) was used to simulate neural dynamics of the 68 cortical ROIs[34]. Based on the simulated neural activity at each ROI, the hemodynamic model[35,36] was then used to simulate blood oxygen level-dependent (BOLD) fMRI. Details of the model can be found in the Methods section. Here we highlight the intuitions behind the MFM. In the MFM, the neural dynamics of each ROI are driven by four components: (1) recurrent (intra-regional) input, (2) inter-regional inputs, (3) external input (potentially from subcortical relays), and (4) neuronal noise. There are "free" parameters associated with each component. First, a larger recurrent connection strength $w$ corresponds to a stronger recurrent input current. Second, the inter-regional inputs depend on the neural activities of other cortical ROIs and the connectional strength between ROIs. The inter-regional connectional strength is parameterized by the SC matrices, scaled by a global scaling constant $G$. Third, $I$ is the external input current. Fourth, the neuronal noise is assumed to be Gaussian with a standard deviation $\sigma$.

In the current study, the recurrent connectional strength $w$, external input current $I$, and noise amplitude $\sigma$ are each parameterized as a linear combination of the principal resting-state FC gradient[27] and T1w/T2w myelin estimate[37], resulting in 10 unknown linear coefficients. Both FC gradient and T1w/T2w map were estimated from the training set. We refer to the resulting model as parametric MFM (pMFM). The 10 unknown linear coefficients were automatically estimated by minimizing disagreement between the empirical and simulated BOLD signal (Fig. 1A).

More specifically, the simulated fMRI was used to compute a $68 \times 68$ static FC matrix and a $1118 \times 1118$ FCD matrix. The agreement between the simulated and empirical static FC matrices was defined as the Pearson's correlation ($r$) between the z-transformed upper triangular entries of the two matrices.

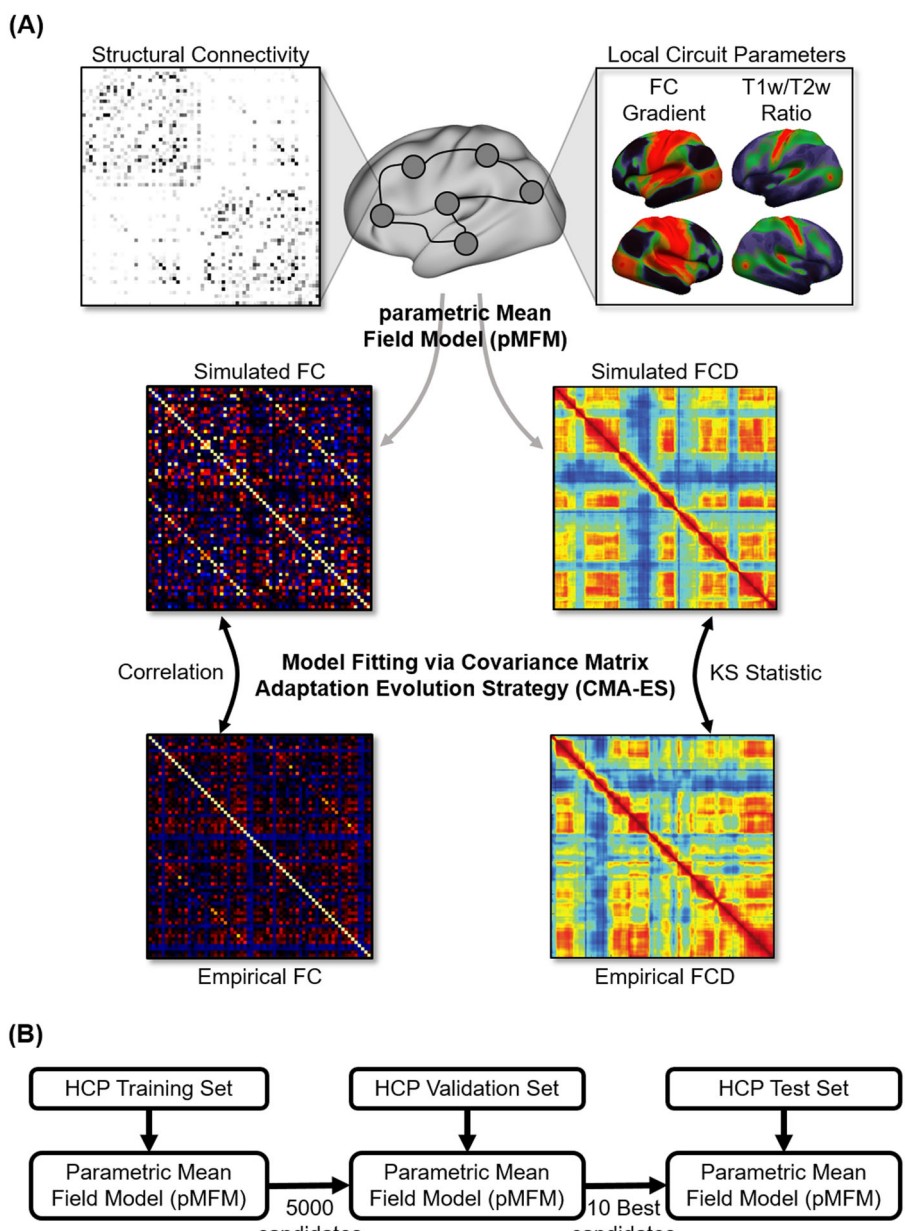

**Fig. 1 Schematic of parametric mean-field model (pMFM) optimization. A** The pMFM comprised ordinary differential equations (ODEs) at each cortical region coupled by a structural connectivity (SC) matrix. The circuit-level parameters were allowed to vary across cortical regions, parameterized by a linear combination of resting-state functional connectivity (FC) gradient and T1w/T2w spatial maps. The pMFM was used to generate simulated static FC and functional connectivity dynamics (FCD). The Covariance Matrix Adaptation Evolution Strategy (CMA-ES) was used to estimate the pMFM by minimizing a cost function of disagreement with empirically observed FC and FCD. **B** The CMA-ES algorithm was applied to the Human Connectome Project (HCP) training set ($N = 351$) to generate 5000 candidate parameter sets. The top 10 candidate parameter sets were then selected from the 5000 candidate sets based on the model fit in the validation set ($N = 350$). Finally, these top 10 candidate sets were evaluated in the HCP test set ($N = 351$). Comparison of the pMFM with other parametrizations (Fig. 3 and Supplementary Fig. S3) utilized the same training-validation-test procedure.

Larger $r$ indicated more similar static FC. The disagreement between the simulated and empirical FCD matrices was defined as the Kolmogorov–Smirnov (KS) distance between the upper triangular entries of the two matrices[22]. A smaller KS distance indicated a more similar FCD. To optimize both static FC and FCD, an overall cost was defined as $(1 - r) + KS$ and minimized in the training set. We considered three different minimization algorithms, each generating 5000 candidate sets of model parameters from the training set. Covariance matrix adaptation evolution strategy (CMA-ES[38]) performed the best in the validation set (Supplementary Fig. S1), so the 10 best CMA-ES

parameter sets from the validation set were evaluated in the test set. The pMFM was compared with other parametrizations using the same training-validation-test procedure.

**The pMFM yielded highly realistic functional connectivity dynamics.** Figure 2A shows a representative empirical FCD from a participant in the test set. Figure 2B shows a simulated FCD generated by the pMFM using the best model parameters (from the validation set) using SC from the test set. Both empirical and simulated FCD exhibited red off-diagonal blocks representing

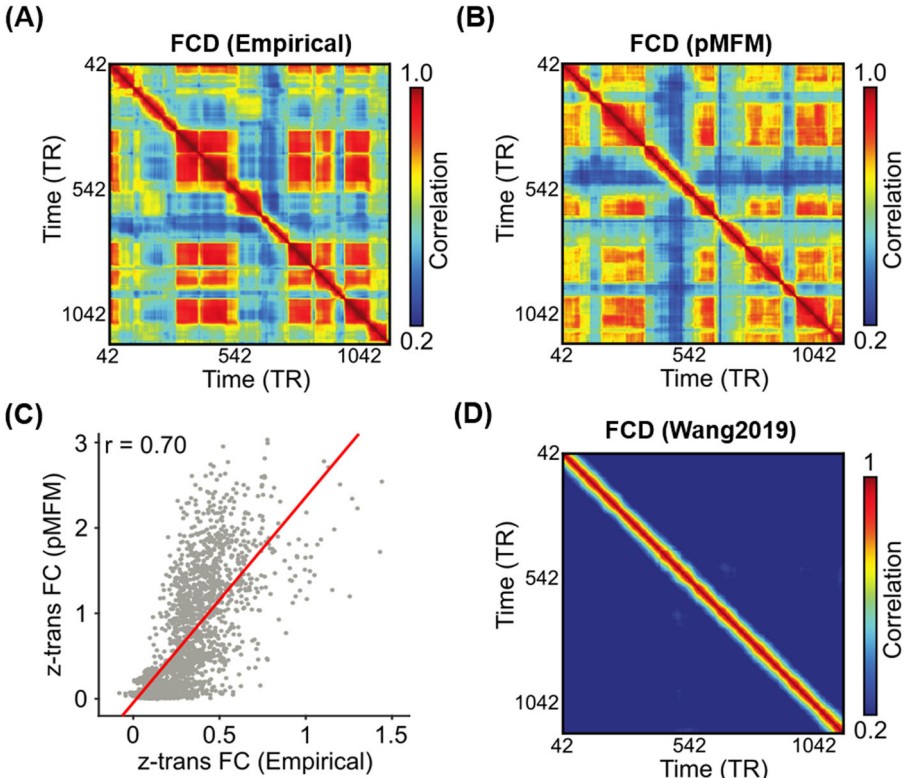

**Fig. 2 Parametric mean-field model (pMFM) generates more realistic static functional connectivity (FC) and functional connectivity dynamics (FCD) than a previous spatially heterogeneous MFM[25]. A** Empirical FCD from a participant from the HCP test set. **B** Simulated FCD from the pMFM using the best model parameters from the validation set using structural connectivity (SC) from the test set. **C** Agreement (Pearson's correlation) between empirically observed and pMFM-simulated static FC using the best model parameters from the validation set using structural connectivity (SC) from the test set. **D** Simulated FCD generated by the previously published spatially heterogeneous MFM[25]. Source data are provided as a Source Data file.

recurring FC patterns. Across the 10 best candidate sets from the validation set, KS distance between empirical and simulated FCD was 0.12 ± 0.03 (mean ± std) in the test set.

Across the 10 best candidate sets from the validation set, correlation between empirical and simulated static FC was 0.66 ± 0.03 in the test set. As a reference, the correlation between SC and static FC in the test set was 0.28. Figure 2C shows the correlation between empirical and pMFM-simulated static FC obtained from applying the best model parameters from the validation set to SC from the test set.

Figure 2D shows the simulated FCD using the MFM parameters from our previous study[25]. The almost constant values in off-diagonal elements suggest a lack of realistic FC dynamics. KS distance between empirical and simulated FCD was 0.88. Correlation between static empirical and simulated static FC was 0.48. Thus, the pMFM was able to generate much more realistic static FC and FCD than the MFM[25].

**Combining anatomical and functional gradients generated more realistic brain dynamics.** In the previous section, we demonstrated that pMFM was able to generate realistic static FC and FCD. To explore what aspects of pMFM are important for generating realistic static FC and FCD, we performed a number of control analyses. First, we investigated the importance of utilizing both anatomical and functional gradients in generating realistic static FC and FCD. Most large-scale circuit model studies assume spatially homogeneous parameters. When recurrent connectional strength $w$, external input current $I$, and noise amplitude $\sigma$ were optimized by CMA-ES, but constrained to be spatially homogeneous (Fig. 3), there was a substantially weaker agreement with empirical static FC ($r = 0.55 \pm 0.05$) and FCD

(KS = 0.50 ± 0.31) in the test set. Similarly, spatial heterogeneity for all three parameters ($w$, $I$, and $\sigma$) were necessary to generate the most realistic static FC and FCD in the test set (Supplementary Fig. S2A–C).

Second, if recurrent connectional strength $w$, external input current $I$, and noise amplitude $\sigma$ were parameterized with only T1w/T2w[24] or only FC gradient, then the resulting static FC and FCD were less realistic in the test set (Fig. 3C). Furthermore, if recurrent connectional strength $w$, external input current $I$, and noise amplitude $\sigma$ were allowed to be spatially heterogeneous across brain regions, but not constrained by T1w/T2w or FC gradient (i.e., non-parametric), then simulations could achieve realistic static FC, but not FCD in the test set (Supplementary Fig. S2D). One reason could be the large number of "free" parameters leading to overfitting in the training set.

Supplementary Table S1 provides summary statistics of the two metrics (FC correlation and FCD KS statistic) in the training, validation, and test sets. The pMFM was statistically better than the spatially homogeneous MFM for both metrics in the test set. Compared with other alternative parameterizations, pMFM was statistically better in one metric and statistically comparable in the other metric.

Finally, instead of fitting to both static FC and FCD in the training set, we also tried fitting only to static FC. Not surprisingly, the resulting model yielded unrealistic functional connectivity dynamics (Supplementary Fig. S3; KS = 0.88 ± 0.01). On the other hand, correlation between static empirical and simulated static FC was 0.73 ± 0.01, which was only slightly better than when optimizing both static FC and FCD (Fig. 2C). This suggests that the goals of generating realistic static FC and FCD were not necessarily contradictory.

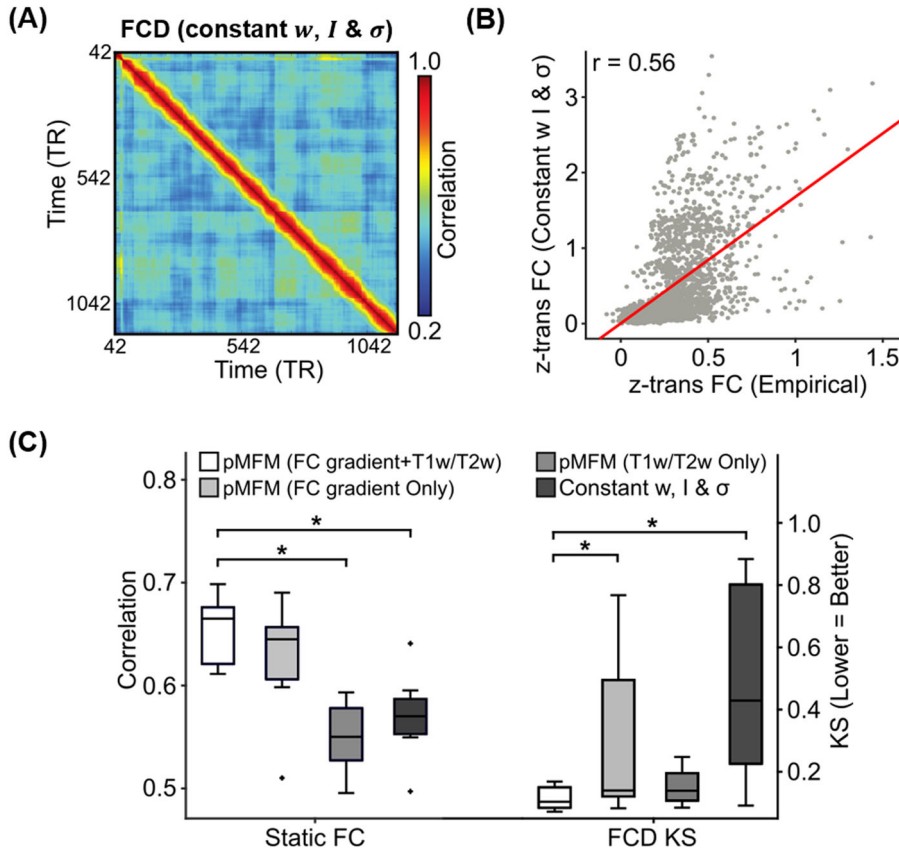

**Fig. 3 Importance of multiple spatial gradients for generating realistic static functional connectivity (FC) and functional connectivity dynamics (FCD).** **A** Simulated FCD from a mean-field model (MFM) optimized using the same algorithm as pMFM, but with model parameters constrained to be the same across cortical regions. **B** Agreement between empirically observed and simulated static FC from MFM optimized using the same algorithm as pMFM, but with model parameters constrained to be the same across cortical regions. **C** Agreement (Pearson's correlation) between simulated and empirically observed static FC, as well as disagreement (KS distance) between simulated and empirically observed FCD across different conditions in the test set. Each boxplot comprises 10 correlation values (left) or 10 KS statistic (right) based on the 10 best candidate sets from the validation set. The boxes show the inter-quartile range (IQR) and the median. Whiskers indicate 1.5 IQR. Black crosses represent outliers. The pMFM utilizing both anatomical and functional gradients (FC gradient and T1w/T2w spatial maps) performed the best, suggesting that T1w/T2w and FC gradient provided complementary contributions. * indicates statistical significance after correcting for multiple comparisons with a false discovery rate (FDR) of $q < 0.05$. All $p$ values are reported in Supplementary Table S1. Source data are provided as a Source Data file.

Overall, these results suggest the importance of parameterizing recurrent connectional strength $w$, external input current $I$, and noise amplitude $\sigma$ with spatial gradients that smoothly varied from sensory-motor to association cortex. Furthermore, T1w/T2w and FC gradient are complementary in the sense that combining the two spatial maps led to more realistic static FC and FCD (Fig. 3).

**Technical considerations and interpretations.** It is worth emphasizing that the different parameterizations were compared with the same training-validation-testing procedure (Fig. 1B), which automatically controls for model complexity or degrees of freedom. A more complex model will generally fit the training data better but might not perform well in the test set. For example, when recurrent connectional strength $w$, external input current $I$, and noise amplitude $\sigma$ were allowed to be spatially heterogeneous across brain regions, but not constrained by T1w/T2w or FC gradient (i.e., non-parametric), then simulations could achieve realistic static FC, but not FCD in test set (Supplementary Table S1). This is an example, where a more flexible model (205 free parameters) yielded an excellent fit in the training and validation sets but a significantly worse fit in the test set.

In all the previous analyses, the overall cost was defined as $(1 − r) + KS$, which placed equal weights on fitting FC and FCD. When the relative weights of FC and FCD were altered, combining both T1w/T2w map and FC gradient still yielded better test set performance than either T1w/T2w map or FC gradient alone (Supplementary Fig. S4). Although the original analysis (Fig. 3) suggested that T1w/T2 map explained FCD better than FC gradient, this was no longer the case when the relative weights were altered (Supplementary Fig. S4). On the other hand, FC gradient was better than T1w/T2w map at explaining static FC across the three experiments (Figs. 3 and Supplementary Fig. S4), which made intuitive sense given that FC gradient was derived from static FC. However, we note that the analyses were not circular given that the FC gradient was derived from the training set and performance was evaluated on the test set.

**Opposite gradient directions in recurrent connection strength, noise amplitude, and external input.** Figure 4B–D illustrates the spatial distribution of recurrent connection strength $w$, external input current $I$, and noise amplitude $\sigma$ based on the best parameter estimate from the validation set. The black lines indicate seven resting-state network boundaries (Fig. 3A[39]). While the

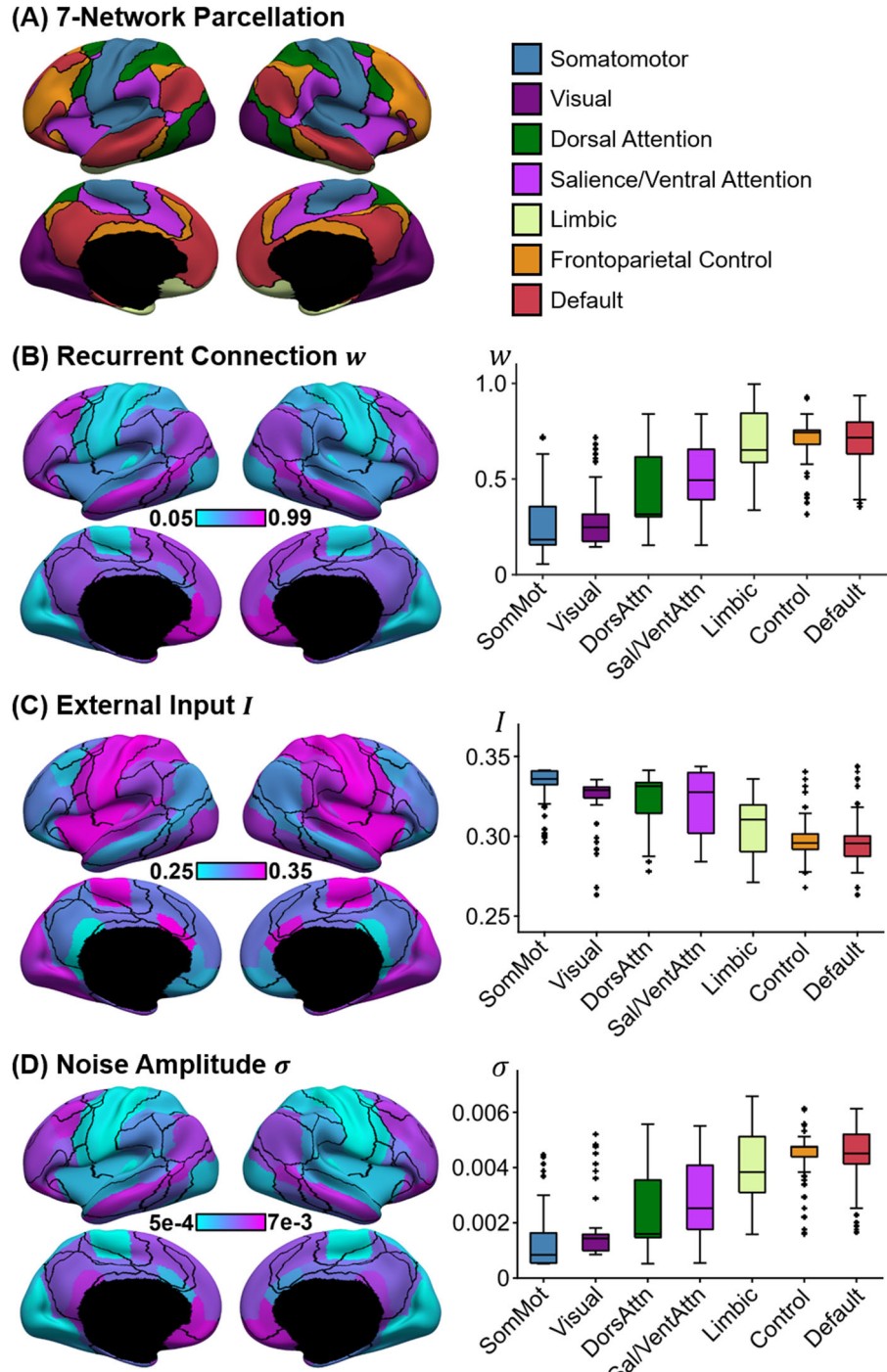

**Fig. 4 Spatial distribution of recurrent connection strength $w$, external input current $I$, and noise amplitude $\sigma$, and their relationships with resting-state networks. A** Seven resting-state networks[39]. **B** Strength of recurrent connection $w$ in 68 Desikan–Killiany cortical ROIs (left) and seven resting-state networks (right). **C** Strength of external input $I$ in 68 Desikan–Killiany cortical ROIs (left) and seven resting-state networks (right). **D** Strength of noise amplitude $\sigma$ in 68 Desikan–Killiany cortical ROIs (left) and seven resting-state networks (right). The boxplots comprised values obtained by "transferring" the parameter estimates from the 68 Desikan–Killiany parcels to all vertices (from the underlying cortical meshes) comprising each anatomical parcel. The vertex wise parameter values were then segregated based on the seven resting-state networks. Therefore, there were 3203, 2478, 1523, 1520, 1067, 1438, and 2886 values comprising the boxplots for somatomotor, visual, dorsal attention, ventral attention, limbic, control, and default networks, respectively. The boxes show the inter-quartile range (IQR) and the median. Whiskers indicate 1.5 IQR. Black crosses represent outliers. Recurrent connection strength and noise amplitude increased from sensory-motor to association (limbic, control, and default) networks. On the other hand, external input current was the highest in sensory-motor networks and decreased towards the default network. Source data are provided as a Source Data file.

resting-state network boundaries do not exactly align with the anatomically defined parcels, there was a striking correspondence between the resting-state networks and estimated pMFM parameters. Given the parameterization of pMFM by a linear combination of FC gradient[27] and T1w/T2w spatial maps[24], it was not surprising that the parameter estimates exhibited a hierarchical gradient of values monotonically changing from sensory-motor to association networks (right column of Fig. 4B–D).

However, the gradient directions were different across the three parameters. In particular, both recurrent connection strength and noise amplitude appeared to increase from sensory-motor to association (limbic, control, and default) networks. On the other hand, external input current was the highest in sensory-motor networks and decreased towards the default network. The directionalities of noise amplitude and external input current were consistent across all the top ten parameter estimates from the validation set. In the case of recurrent connection strength, one of the ten parameter sets exhibited the opposite direction (i.e., decrease from sensory-motor regions to association networks; Supplementary Fig. S5), suggesting potential degeneracy in the case of recurrent connection strength.

The previous analysis was "biased" to find degeneracy given that the top 10 parameter sets were selected to ensure diversity (see "Methods" section). To further explore the degeneracy issue, the recurrent connection strength map of the top parameter set (from the validation set) was correlated with the recurrent connection strength maps of the remaining 4999 candidate parameter sets (Supplementary Fig. S6). In general, parameter sets with good validation cost were strongly correlated with the top parameter estimate from the validation set. Similar conclusions were obtained for external input and noise amplitude, although external input appeared to be less stable than recurrent connection strength and noise amplitude.

**Time-varying amplitude of regional fMRI time courses tracks time-varying functional connectivity**. Given that the pMFM was able to generate realistic FCD, we now seek to use the pMFM to provide further insights into mechanisms underlying FCD. Previous studies have suggested that FCD might arise from switching between multi-stable states[22,29]. Indeed, a magnified portion of the FCD matrix from a HCP test participant (Fig. 5A) suggests the presence of at least two distinct states. In one state (white asterisk in Fig. 5A), the sliding window FC pattern appeared to be coherent over a time period. In a second state (black asterisk in Fig. 5A), the sliding window FC patterns were incoherent over another time period, so the high correlations within the block were restricted to the diagonals, and likely driven by autocorrelation in the fMRI signals and overlapping sliding windows. We hypothesized that fMRI signals might be dominated by large coherent amplitude fluctuations during the coherent state and dominated by noise during the incoherent state (right panel in Fig. 5A; see ref. [40] for a review of multi-stability). If our hypothesis were true, we would expect large regional fMRI signal amplitude during the coherent state and small regional fMRI signal amplitude during the incoherent state.

To test our hypothesis, the standard deviation of the average fMRI signal of each cortical ROI within each sliding window was computed. Figure 5B (top panel) shows the FCD matrix of a single participant from the HCP test set. Figure 5C (top panel) shows the simulated FCD matrix from the pMFM using the best model parameters from the validation set and structural connectivity (SC) from the test set. The middle panels of Fig. 5B, C show the FCD mean time course obtained by averaging the rows of the FCD matrices from the top panels. Sharp transitions in the FCD mean time course reflected sharp transitions in the

FCD matrix. The bottom panel shows the sliding window standard deviation (SW-STD) of empirical and simulated fMRI signals. There was striking correspondence between sharp transitions in the FCD mean time course and SW-STD time courses in both empirical and simulated data (red dashed lines in Fig. 5B, C).

Consistent with our hypothesis, there was a large signal amplitude during the coherent state and low signal amplitude during the incoherent state (Fig. 5B). To quantify this phenomenon, for each run of each participant in the HCP test set, we fitted a mixture of two Gaussian distributions to the histogram of the FCD mean[28]. The cross-over point of the two Gaussian distributions was used to threshold the FCD mean. Time points with FCD mean greater than the threshold were designated as the coherent state (high FCD mean), while time points with FCD mean lower than the threshold were designated as the incoherent state (low FCD mean). The SW-STD was then averaged across all cortical regions and all runs of each participant. As shown in Fig. 5D, the SW-STD was significantly higher during the coherent state than the incoherent state ($p = 6.9e-150$). Similar results were obtained for the pMFM simulations (Fig. 5E). The dwell time distributions of the two states were also similar between the empirical and simulated data (Supplementary Fig. S7). The two distributions appeared to follow an exponential distribution (as opposed to a Gamma distribution), suggesting the presence of multi-stability rather than meta-stability[40].

**Sensory-motor regions drive switching behavior in functional connectivity dynamics**. In the previous section, we found a striking correspondence between the FCD mean time course and the regional SW-STD time courses (Fig. 5B, C). We note that the FCD mean time course reflected cortex-wide fluctuations in FC patterns, while SW-STD time courses were region-specific. Therefore, to investigate regional heterogeneity of FCD-STD correspondence (Fig. 5) across the cortex, the correlation between the first derivative of the FCD mean time course and the first derivative of the SW-STD time course was computed for each cortical region. In the case of empirical observations, the FCD-STD correlations were averaged across all runs of all participants in the test set yielding a final FCD-STD correlational spatial map (Fig. 6A). In the case of pMFM simulations, the correlations were averaged across 1000 random simulations using the best model parameters from the validation set using structural connectivity (SC) from the test set, yielding a final FCD-STD correlational spatial map (Fig. 6B).

Statistical significance was established using a permutation test (see "Methods" section). Almost all cortical regions were significant after correcting for multiple comparisons (FDR $q <$ 0.05; Fig. S8). Across both pMFM simulations and empirically observed data, FCD-STD correlations were the highest in sensory-motor regions and lowest in the association cortex. There was strong spatial correspondence between simulated and empirical results ($r = 0.87$; Fig. 6C). We note that the pMFM was optimized to yield realistic FCD with no regard for spatial correspondence, so the high level of spatial correspondence suggests that the pMFM was able to generalize to new unseen properties of FCD.

To explore the causal relationship between sensory-motor regions and FCD, we tested whether perturbation of sensory-motor regions could "kick" the system from an incoherent FCD state to a coherent FCD state. Among 1000 random simulations of pMFM, time segments in the incoherent state (low FCD mean) lasting for at least 200 contiguous fMRI time points were selected. The neural signals of the top five FCD-STD regions (sensory-motor drivers; Fig. 6B)

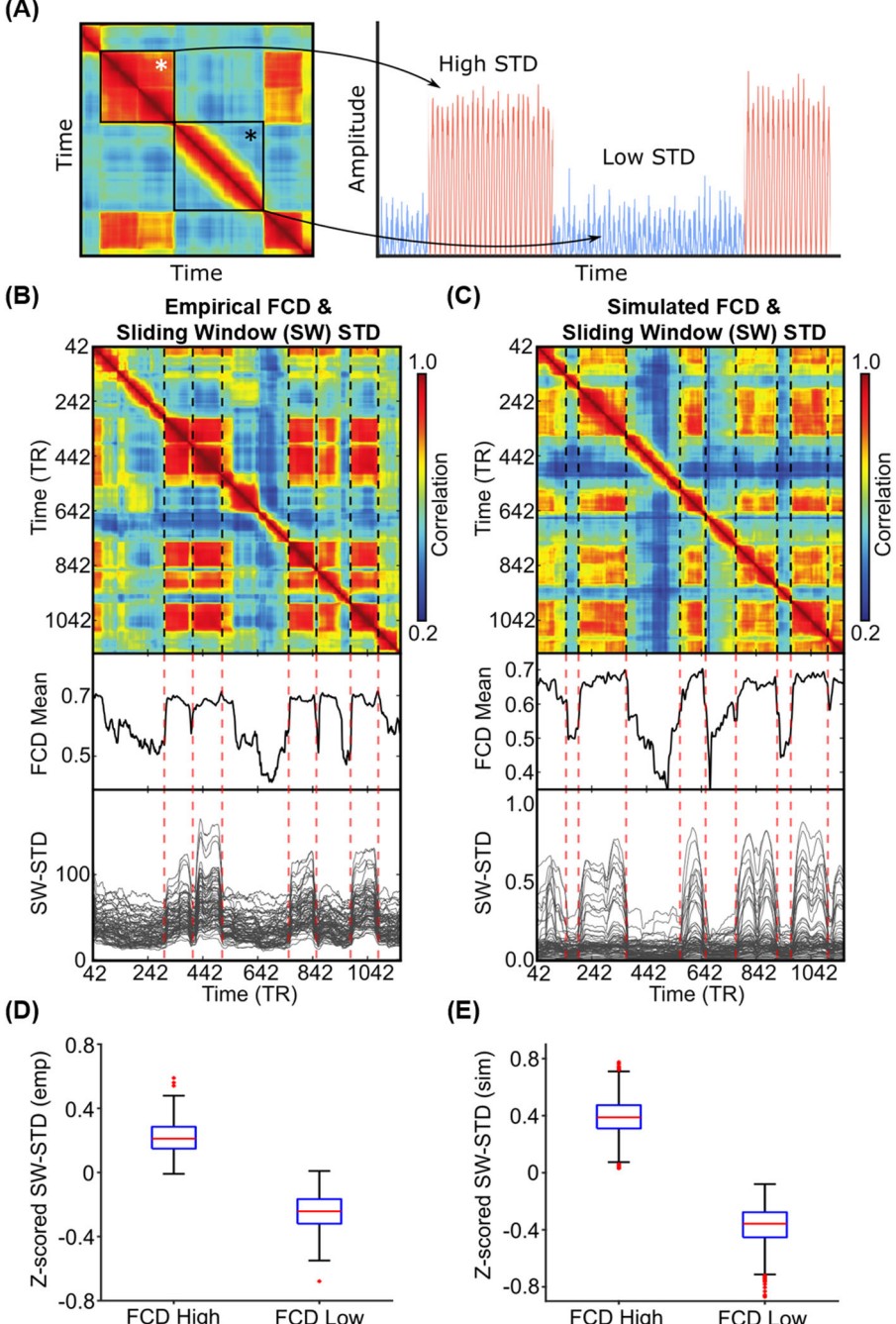

**Fig. 5 Correspondence between functional connectivity dynamics (FCD) and time-varying amplitude of regional fMRI time courses. A** Inspection of FCD from a HCP test participant suggests at least two states. The first state (white asterisk) exhibits coherent FC patterns over a period of time. The second state (black asterisk) exhibits incoherent FC patterns over a period of time. The right panel illustrates our hypothesis that the coherent state might be characterized by large coherent amplitude in regional fMRI signals, i.e., high standard deviation (STD), while the incoherent state might be characterized by noise in regional fMRI signals, i.e., low standard deviation (STD). **B** Top panel shows empirical FCD matrix of a HCP test participant. The middle panel shows the FCD mean time course obtained by averaging the rows of the FCD matrix from the top panel. The bottom panel shows the standard deviation of each regional fMRI time course within each sliding window (SW-STD). The color of the lines corresponds to the correlation between the first derivative of the FCD mean time course and the first derivative of the SW-STD time courses. Sharp transitions in SW-STD corresponded to sharp FCD transitions (red dashed lines). **C** Same as **B**, but simulated from pMFM using the best model parameters from the validation set and structural connectivity from the test set. **D** SW-STD during coherent (high FCD mean) and incoherent (low FCD mean) states. Boxplots illustrate the variation across HCP test participants. Coherent states were characterized by large amplitude (STD) in fMRI signals ($p = 2.4e−168$). The $p$-value was computed from a two-sided $t$-test and survived the false discovery rate ($q < 0.05$). **E** Same as **D**, but simulated from pMFM. There are 349 and 1000 independent samples for the boxplots in **D** and **E**, respectively. The boxes show the inter-quartile range (IQR) and the median. Whiskers indicate 1.5 IQR. Red crosses represent outliers. Source data are provided as a Source Data file.

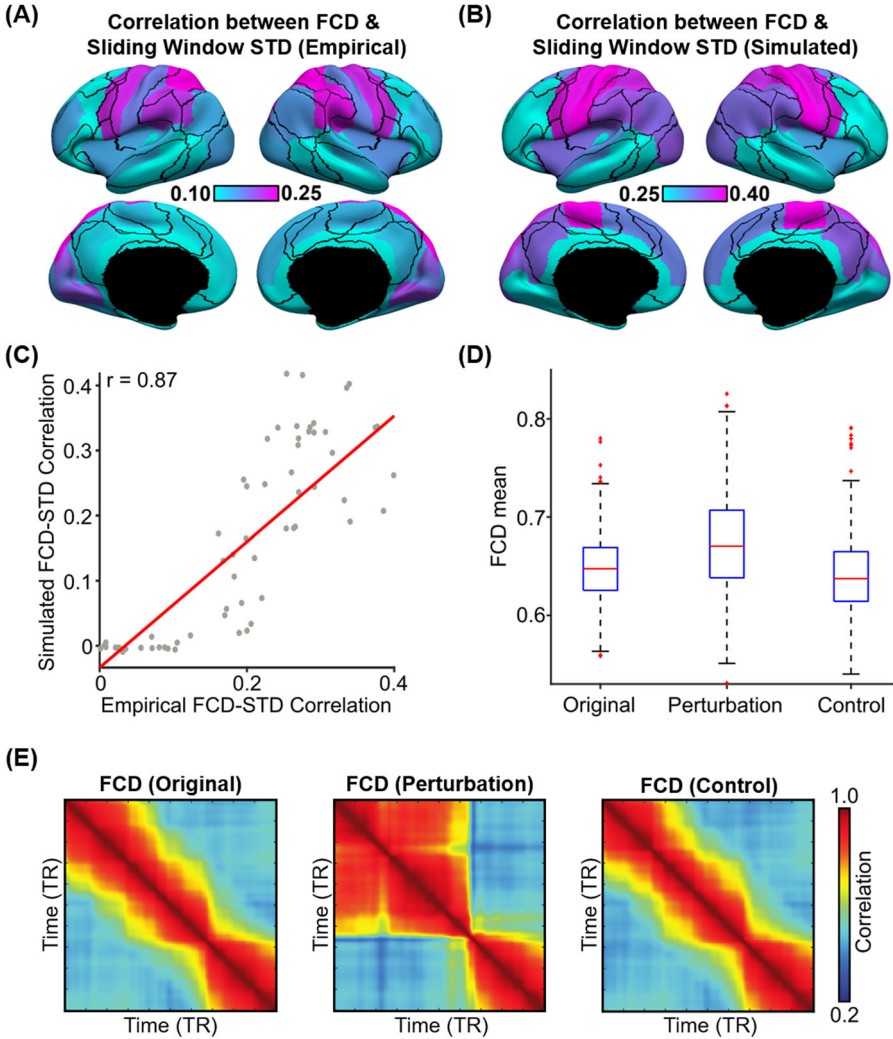

**Fig. 6 Sensory-motor regions drive sharp transitions in functional connectivity dynamics (FCD). A** FCD-STD correlations obtained by correlating the first derivative of the FCD mean time course and the first derivative of the SW-STD time course of each cortical region. These correlations were performed for each HCP test participant and averaged across all runs and participants. **B** Same as **A** but simulated from pMFM using the best model parameters from the validation set and structural connectivity from the test set. The correlations were averaged across 1000 random simulations. **C** Correlation between empirical and simulated FCD-STD correlation spatial maps from **B** and **C**, showing strong correspondence between empirical and simulated results.
**D** Casual perturbation of top 5 FCD-STD correlated regions (**B**) during the incoherent state (low FCD mean) led to a transition into the coherent state (high FCD mean). As a control analysis, perturbation of the bottom 5 FCD-STD correlated regions (**B**) during the incoherent state (low FCD mean) did not lead to a state change (FCD mean remains low). There are 297 independent samples for each boxplot in **D**. The boxes show the inter-quartile range (IQR) and the median. Whiskers indicate 1.5 IQR. Red crosses represent outliers. **E** Example FCD from the perturbation experiments. (Left) original incoherent state. (Middle) perturbation of top 5 FCD-STD correlated regions (sensory-motor drivers). (Right) perturbation of bottom 5 FCD-STD correlated regions. Source data are provided as a Source Data file.

were then perturbed to increase their amplitude. The perturbation led to the successful transition of the FCD into a more coherent state with a higher FCD mean ($p = 6e{-}14$; Fig. 6D). Perturbation of the bottom five FCD-STD regions (Fig. 6B) did not lead to an increase in FCD mean. Figure 6E illustrates the example results of the perturbation experiment. Similar results were obtained if we perturbed the top 10 and bottom 10 regions. Overall, this suggests that sensory-motor regions were a driver of switching behavior in FCD.

**Parvalbumin–somatostatin and first genetic principal component correlate with sensory-motor drivers of time-varying functional connectivity dynamics.** Results from the previous sections suggest that time-varying amplitude of sensory-motor regions tracks switching behavior in time-varying functional

connectivity. A recent study[30] demonstrated that the difference in the spatial distribution of molecular markers of parvalbumin and somatostatin interneurons (PVALB-SST) is linked with the amplitude of regional fMRI signals (Fig. 7A). This intriguing finding is in line with data in rodents documenting the importance of these interneuron classes in local cortical circuit function[31]. Inspection of the cortical distribution of PVALB-SST transcripts from the Allen Human Brain Atlas (AHBA) dataset (Fig. 7A) suggests a strong similarity with the FCD-STD correlational spatial maps (Fig. 6).

PVALB-SST (Fig. 7A) was averaged within each cortical ROI and correlated with the FCD-STD correlational spatial maps (Fig. 6). The correlations were 0.72 and 0.65 for the empirical (Fig. 7B) and simulated (Fig. 7C) data, respectively. As shown in Fig. 7D, both correlations were significant based on spin-tests preserving spatial autocorrelation[41,42]. To test for specificity of

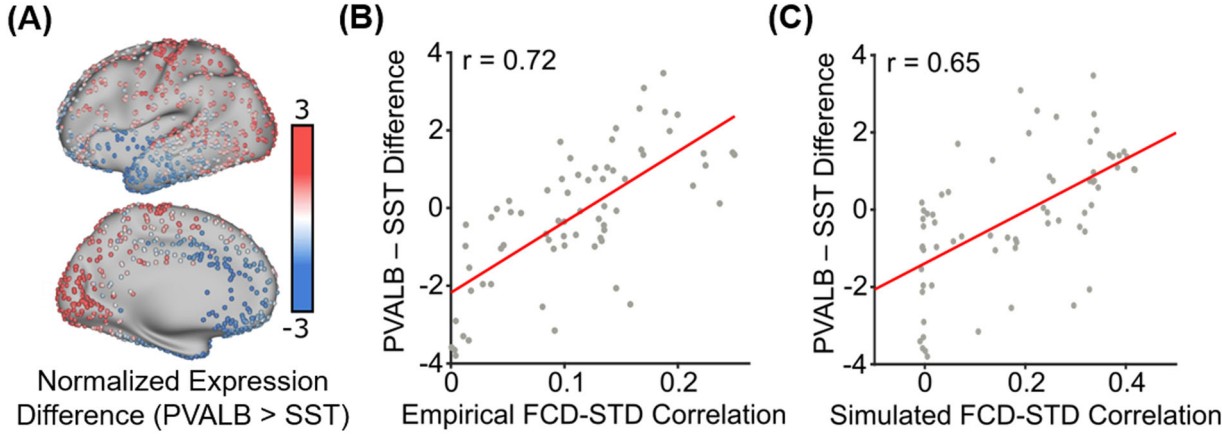

**Fig. 7 Correlations between the spatial distribution of sensory-motor drivers (FCD-STD correlational spatial maps) and gene expression spatial maps. A** Difference in normalized expressions of parvalbumin and somatostatin (PVALB-SST) from the Allen Human Brain Atlas (AHBA). Panel is a re-rendering of ref. [30]. **B** Correlation between empirical FCD-STD correlational map (Fig. 6B) and PVALB-SST gene expression map. **C** Correlation between simulated FCD-STD correlational map (Fig. 6C) and PVALB/SST gene expression map. **D** Table of correlations between FCD-STD correlational spatial maps and two gene expression maps: PVALB-SST and first principal component of gene expression[26,43]. The "spin test" tested the significance of the correlations while controlling for spatial autocorrelation. The "random gene pair" tested for the specificity of PVALB-SST by randomly sampling pairs of brain-specific genes. *P*-values that survived the false discovery rate (*q* < 0.05) are bolded. Standard deviations reported in the table were obtained by bootstrapping. Source data are provided as a Source Data file.

PVALB-SST, a null distribution was also generated based on random pairs of brain-specific genes. Both correlations were again significant (Fig. 7D). Overall, this suggests that the spatial distribution of sensory-motor drivers was associated with the differential expression of PVALB and SST

Given that previous studies have suggested the existence of multiple similar gene expression gradients, the first principal component of AHBA brain-specific gene expression data[26,43] was correlated with the FCD-STD correlational spatial maps (Fig. 6). The first gene expression principal component was also correlated with both empirical and simulated FCD-STD spatial maps, although the correlations were slightly weaker than the correlations with the PVALB-SST gene expression map (Fig. 7D).

The recurrent connection strength *w* and noise amplitude *σ* were also correlated with the PVALB-SST gene expression map under the spin-test, but not the random-gene-pair tests. This suggests a lack of specificity to PVALB-SST (Fig. 7D). The external input *I* was not correlated with any gene expression pattern.

**Specificity of T1w/T2w map and FC gradient**. We have shown that combining T1w/T2w map and FC gradient led to more realistic brain dynamics than using either no gradient or only one gradient (Fig. 3). To further explore the specificity of the parameterization, we repeated the training-validation-test procedure (Fig. 1B) using randomly rotated versions of T1w/T2w map and/or FC gradient. Despite having the same degrees of freedom as the original pMFM, the rotated parameterizations led to a worse fit to static FC and/or FCD in the test set (Supplementary Fig. S9).

We also repeated the training-validation-test procedure with alternate gradient maps, including the second FC gradient[27], inter-subject functional connectivity variability map[44], first structural covariance gradient[45], and the first genetic principal component (Supplementary Fig. S10). To provide additional context, Supplementary Fig. S10 shows the correlations among the different gradient maps and the top estimated model parameters (*w*, *I*, *σ*) from the original pMFM (Fig. 4).

The estimated model parameters were most strongly correlated with the first principal gradient, although we note that the first

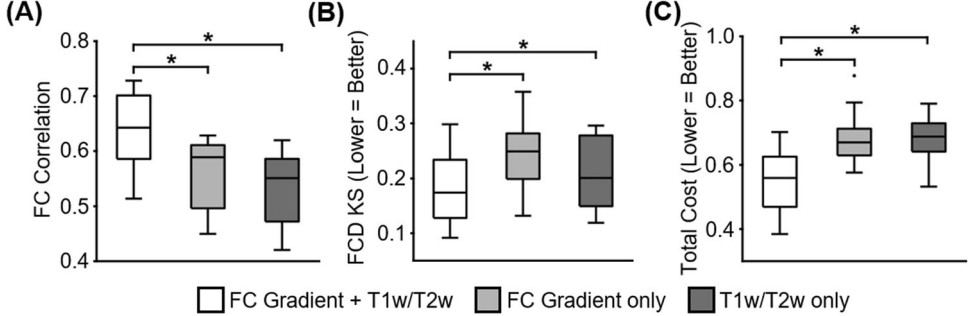

**Fig. 8 At the individual-level, pMFM parameterized by both group-level FC gradient and T1w/T2w map yielded more realistic static FC and FCD than FC gradient or T1w/T2w map alone. A** Agreement (Pearson's correlation) between simulated and empirically observed static FC in the test sessions of individual participants. **B** Disagreement (KS distance) between simulated and empirically observed FCD in the test sessions of individual participants. **C** Total cost in the test sessions of individual participants. Each boxplot comprises 12 FC correlation values, 12 FCD KS or 12 total cost values of the 12 individual participants. The boxes show the inter-quartile range (IQR) and the median. Whiskers indicate 1.5 IQR. Black crosses represent outliers. *Indicates statistical significance after correcting for multiple comparisons with a false discovery rate of $q < 0.05$. Source data are provided as a Source Data file.

principal gradient alone did not lead to the best performance in the test set (Supplementary Fig. S10). Instead, the best single parameterization was the T1w/T2w map. Combining T1w/T2w map with the first FC gradient (i.e., original pMFM) led to the best performance in the test set, but the improvement was not statistically significant when the T1w/T2w map was replaced with inter-subject FC variability, the first genetic principal component or second FC gradient. However, we note that in these cases, the resulting FCD-STD correlation maps remained highly similar to the original FCD-STD map (Fig. 6B) with correlations >0.9, suggesting that these cortical features may index similar underlying mechanisms.

**Control analyses**. To ensure the robustness of results, we performed several control analyses. First, we note that the simulation of pMFM utilized 10 ms time step. To ensure that this time step was sufficiently small, the best model parameters from the validation set were applied to the test set using 1 ms time step. KS distance between empirical and simulated FCD in the test set was 0.11 ± 0.05. The correlation between empirical and simulated static FC was 0.66 ± 0.03.

Second, the previous analyses utilized a sliding window comprising 83 time points for computing FCD. To ensure the model parameters generalized to different window lengths, empirical and simulated FCD was computed in the test set using window lengths of 43 and 125. KS distance between empirical and simulated FCD in the test set was 0.15 ± 0.07 and 0.14 ± 0.04 for window lengths 43 and 125, respectively.

Third, we investigated whether the FCD-STD correlation maps (Fig. 6) might be influenced by global signal fluctuation. We repeated the analysis by restricting to 50 test participants with the lowest global signal fluctuation. The resulting FCD-STD correlation map was very similar to the original results ($r = 0.82$).

Fourth, although time-varying FC was represented using the FCD matrix (Fig. 2A), other representations could be possible. Zalesky and colleagues explored time-varying FC by computing time-varying network efficiency for each sliding window[13]. They found high and low-efficiency states, which appeared to correspond to the high and low coherent states in the FCD matrix (Supplementary Fig. S12A). The pMFM also captured these high and low-efficiency states in test set (Supplementary Fig. S12B). On the other hand, the spatially homogeneous MFM could generate high and low-efficiency states in the training set, but not the test set (Supplementary Fig. S12C, D).

Fifth, we replicated our results with a higher resolution parcellation with 100 cortical ROIs[46]. Consistent with our main results, we found that pMFM yielded more realistic simulated FC and FCD in the test set (Supplementary Fig. S13) compared with our previous study[25]. Across all 10 best parameter sets from the validation set, noise amplitude increased from sensory-motor to association (limbic, control, and default) networks, while external input exhibited the opposite direction. In 8 of the 10 best parameter sets, recurrent connect strength increased from sensory-motor to association (limbic, control, and default) networks, thus again suggesting potential degeneracy (Supplementary Fig. S14).

In the Schaefer parcellation, time-varying amplitude of sensory-motor time courses tracks switching behavior in time-varying functional connectivity (Supplementary Figs. S15 and S16). Causal perturbation analysis also confirmed that sensory-motor regions appeared to drive transitions in FCD (Supplementary Fig. S16). Both simulated and empirical FCD-STD correlation maps were correlated with PVALB-SST gene expression maps (Supplementary Table S2). Both correlations were significant under the spin-test and random gene-pair tests. The simulated, but not the empirical, FCD-STD correlation maps were correlated with the first principal component of gene expression.

Finally, to explore the possibility of individual-level pMFMs, we considered 12 participants from the HCP test-retest dataset that overlapped with our test set. There were four MRI sessions for each participant. The first two sessions and the last two sessions were on average 3.8 ± 1.5 months apart. Similar to previous analyses, the pMFM was optimized using group-level FC gradient and group-level T1w/T2w map from the training set. The main difference is that the model was optimized using group-level SC from the test set, as well as static FC and FCD from the first two sessions of individual participants. The top 10 parameter sets from the first two sessions were then evaluated in the remaining two sessions. We found that combining T1w/T2w map and FC gradient yielded more realistic static FC and FCD than using T1w/T2w map or FC gradient alone at the individual level (Fig. 8). Future studies will explore whether individual-level FC gradient, T1w/T2w map, and SC could bring further benefits to individual-level MFMs.

## Discussion
By incorporating anatomical and functional gradients into the parameterization of local circuit properties, the resulting large-scale circuit model generated realistic time-averaged (static) and

time-varying (dynamic) properties of large-scale spontaneous brain activity. Both empirical and simulated fMRI data exhibited multi-stable properties, in which there was spontaneous switching between a high coherent state and a low coherent state. The multi-stability was tracked by the time-varying amplitude of regional fMRI signals. By performing causal perturbations of the large-scale circuit model, we demonstrated that spontaneous amplitude fluctuations of sensory-motor regions were a driver of the observed switching behavior. Furthermore, the relationship between regional fMRI amplitude and functional connectivity dynamics was also associated with PVALB-SST and the first principal component of gene expression, suggesting that heterogeneity in excitation-inhibition balance might shape multi-stability in FC dynamics.

**Anatomical and functional gradients contribute to spontaneous brain dynamics**. Previous studies have proposed a dominant gradient of cortical organization with sensory-motor and association regions at opposing ends[47]. Supporting this idea of a dominant axis, many studies have emphasized similarities among gradients estimated from diverse sources, including resting-state FC principal gradient, T1w/T2w myelin estimate, gene expression data, functional task activation, and computational modeling[25–27,48,49]. Yet, there are clear differences among the gradients and a growing number of studies have suggested dissociations among multiple spatially similar gradients[45,50,51]. Here, we showed that by parameterizing local circuit parameters with both anatomical (T1w/T2w) and functional (FC) gradients, the resulting mean-field model was able to generate dramatically more realistic static FC and FC dynamics than either gradient alone (Fig. 3).

Our control analysis with alternate gradient maps suggests that combining the T1w/T2w map with the first FC gradient led to the best performance, but T1w/T2w could be replaced with inter-subject FC variability, first genetic principal component, or second FC gradient without statistically significant loss in performance. Furthermore, while it made intuitive sense that utilizing the resting-state FC gradient would help to explain resting-fMRI dynamics, the training-validation-test scheme ensured the analysis was not circular.

The optimized mean-field model exhibited opposing gradient directions across local circuit parameters (Fig. 4). Across all top ten parameter sets, noise amplitude increased from sensory-motor to association cortex, while external input decreased from sensory-motor to association cortex. The higher external input in sensory-motor regions might reflect the flow of sensory information from the external environment via subcortical relays. In the case of the recurrent connection strength, nine of the ten best parameter sets exhibited increasing values from sensory-motor to association cortex, but one parameter set exhibited the opposite direction. Thus, recurrent connection strength might exhibit potential degeneracies in mean-field models, thus explaining contradictions in the literature[24,25].

**Multi-stability in spontaneous brain dynamics**. The spontaneous ebb and flow observed in FC dynamics is an intriguing property that has fascinated the field[12,14,22,33,52,53]. As shown in Fig. 5A, there are periods of brain activity with strong coherent FC and periods with incoherent FC. We found that the coherent FC state was characterized by larger fMRI signal amplitude across brain regions, while the incoherent FC state was characterized by smaller fMRI signal amplitude (Fig. 5). Intriguingly, transitions in the regional amplitude of sensory-motor regions appeared to track switching behavior in FC dynamics (Fig. 6). Perturbations

of the mean-field model suggest that this relationship might be causal.

Regional fMRI amplitude has been previously linked with the differential expression of PVALB and SST across the cortex[30]. PVALB and SST interneurons preferentially target perisomatic regions and dendrites of pyramidal cells, respectively, and are thought to regulate synaptic outputs and inputs, respectively[54]. Thus the spatially heterogeneous distribution of PVALB and SST interneurons[31] might modulate regional neural signal amplitude[30]. Here, we found that PVALB-SST gene expression map correlates with the spatial distribution of sensory-motor drivers whose time-varying amplitude tracks functional connectivity dynamics (Fig. 7).

However, we note that this association cannot be solely attributed to PVALB-SST given that the gradients of PVALB-SST expression are embedded within a broader pattern of gene expression variation across the cortex[26,43]. Indeed, the spatial distribution of sensory-motor drivers was also correlated with the first principal component of cortical genes (Fig. 7). The first gene principal component has been shown to strongly correlate with the spatial distribution of genes coding for different excitatory and inhibitory neurons[26], which might reflect spatial heterogeneity in excitation-inhibition balance[55]. Overall, this suggests a potential link between FC dynamics and heterogeneity in excitation/inhibition balance across the cortex.

## Methods

**Data**. We considered 1052 participants from the Human Connectome Project (HCP) S1200 release[56]. All participants were scanned on a customized Siemens 3T Skyra using a multi-band sequence. Four resting-state fMRI (rs-fMRI) runs were collected for each participant in two sessions on two different days. Each rs-fMRI run was acquired with a repetition time (TR) of 0.72 s at 2 mm isotropic resolution and lasted for 14.4 min. The diffusion imaging consisted of 6 runs, each lasting ~9 min and 50 s. Diffusion weighting consisted of 3 shells of $b = 1000$, 2000, and 3000 s/mm$^2$ with an approximately equal number of weighting directions on each shell. Details of the data collection can be found elsewhere[56]. The 1052 subjects were randomly divided into training ($N = 351$), validation ($N = 350$) and test ($N = 351$) sets.

Data collection was approved by a consortium of institutions institutional review boards (IRBs) in the United States and Europe, led by Washington University (St Louis) and the University of Minnesota (WU-Minn HCP Consortium). The current study was approved by the IRB of the National University of Singapore.

**Preprocessing**. Details of the HCP preprocessing can be found in the HCP S1200 manual. We utilized rs-fMRI data, which had already been projected to fsLR surface space, denoised with ICA-FIX and smoothed by 2 mm. For each run of each participant, the fMRI data were averaged within each Desikan–Killiany[32] ROI to generate a $68 \times 1200$ matrix. Each $68 \times 1200$ matrix was used to compute $68 \times 68$ FC matrix by correlating the time courses among all pairs of time courses. The FC matrices were then averaged across runs of participants within the training (or validation or test) set, resulting in a group-averaged training (or validation or test) FC matrix.

Functional connectivity dynamics (FCD) was computed as follows. For each run of each participant, FC was computed within each of 1118 sliding windows. The length of each sliding window was 83 time points (60 s) as recommended by previous studies[33,57]. We note that our results were robust to window length (see "Control analysis" in the Results section). Each sliding window FC matrix was then vectorized by only considering the upper triangular entries. The vectorized FCs were correlated with each other generating a $1118 \times 1118$ FCD matrix.

In the case of diffusion MRI, generalized Q-sampling imaging (GQI) was used to reconstruct the white matter pathways, allowing for complex diffusion fiber configurations and streamline tractography[58]. A $68 \times 68$ structural connectivity (SC) matrix was generated for each subject, where each entry corresponded to the number of streamlines between two ROIs. To generate a group-level SC matrix, a thresholding procedure was employed to remove false positives. More specifically, if <50% of participants had a non-zero value in a particular entry in the SC matrix, then the entry is set to zero in all individual-level SC matrices. For each SC entry, the number of streamlines was averaged across participants with non-zero streamlines. Separate group-level SC matrices were computed for the training, validation, and test sets.

**Dynamic mean-field model (MFM)**. The MFM was derived by the mean-field reduction of a detailed spiking neuronal network model[34]. For each cortical ROI,

the neural activity obeys the following nonlinear stochastic differential equations:

$$\dot{S}_i = -\frac{S_i}{\tau_s} + r(1 - S_i)H(x_i) + \sigma v_i(t) \qquad (1)$$

$$H(x_i) = \frac{ax_i - b}{1 - \exp(-d(ax_i - b))} \qquad (2)$$

$$x_i = wJS_i + GJ\sum_j C_{ij}S_j + I, \qquad (3)$$

where $S_i$, $H(x_i)$, and $x_i$ denote the average synaptic gating variable, population firing rate, and total input current of the $i$th cortical ROI. The total input current $x_i$ is the superposition of three inputs. The first input, the intra-regional input, is controlled by the recurrent connection strength $w$. The second input, the inter-regional input, is controlled by the SC matrix ($C_{ij}$ is the SC between regions $i$ and $j$), as well as a global scaling factor $G$. The third input is the external input current $I$, which might include inputs from subcortical relays. Following previous studies[25,34], the synaptic coupling $J$ was set to 0.2609 (nA). The parameter values of the input-output function $H(x_i)$ were set to $a = 270(n/C)$, $b = 108$(Hz), and $d = 0.154$(s). The kinetic parameters for synaptic activity were set to $r = 0.641$ and $\tau_s = 0.1$(s). $v_i(t)$ is uncorrelated standard Gaussian noise and the noise amplitude is controlled by $\sigma$.

The simulated neural activities $S_i$ were fed to the Balloon–Windkessel hemodynamic model[35,36] to simulate the fMRI BOLD signals for each ROI. The equations and parameters were exactly the same as our previous study[25]. More specifically, the MFM and hemodynamic model were simulated using Euler's integration with a time step of 10 ms. The starting values of $S_i$ in the MFM were randomly initialized. Simulation length for the fMRI signals was 16.4 min. The first 2 min of the fMRI signals were discarded and the time series were downsampled to 0.72 s to have the same temporal resolution as the empirical fMRI signals in the HCP. The simulated fMRI signals could then be used to generate simulated FC and FCD matrices.

**Parametric mean-field model (pMFM)**. In our previous study[25], the recurrent connection strength $w$, external input current $I$, global constant $G$ and noise amplitude $\sigma$ were optimized by fitting to static FC. The recurrent connection strength $w$ and external input current $I$ were allowed to vary independently across cortical ROIs, while $G$ and $\sigma$ were assumed to be constant. On the other hand[24], parameterized the recurrent connection strengths with the T1w/T2w myelin map.

In this study, recurrent connection strength $w$, external input current $I$ and noise amplitude $\sigma$ were allowed to vary across brain regions, while $G$ was kept as a constant. Instead of allowing $w$, $I$ and $\sigma$ to vary independently[25], we parameterized $w$, $I$ and $\sigma$ as linear combinations of group-level T1w/T2w myelin maps[37] and the first principal gradient of functional connectivity[27]:

$$w_i = a_w\mathbf{Mye}_i + b_w\mathbf{Grad}_i + c_w \qquad (4)$$

$$I_i = a_I\mathbf{Mye}_i + b_I\mathbf{Grad}_i + c_I \qquad (5)$$

$$\sigma_i = a_\sigma\mathbf{Mye}_i + b_\sigma\mathbf{Grad}_i + c_\sigma, \qquad (6)$$

where $w_i$, $I_i$, and $\sigma_i$ denoted the recurrent connection strength, external input current, and noise amplitude, respectively, of the $i$th cortical region. $\mathbf{Mye}_i$ and $\mathbf{Grad}_i$ were the average values of the T1w/T2w myelin map and the first FC principal gradient within the $i$th cortical ROI. Both T1w/T2w myelin maps and first principal gradient of functional connectivity were computed from the HCP training set. Therefore, there are a total of 10 unknown parameters: $G$ and linear coefficients ($a_w, b_w, c_w, a_I, b_I, c_I, a_\sigma, b_\sigma, c_\sigma$). These unknown parameters were be estimated from the HCP training set (see next section).

**Cost function to minimize disagreement with empirical static FC and FCD**. The 10 unknown parameters in the pMFM were estimated by maximizing fit to static FC and FCD in the HCP training set. For a particular set of parameters, the pMFM could be used to generate simulated FC and FCD matrices. The agreement between the simulated and empirical static FC matrices was defined as the Pearson's correlation ($r$) between the $z$-transformed upper triangular entries of the two matrices. Larger $r$ indicated more similar static FC. Pearson's correlation was chosen given its popularity in the literature. However, we note that Pearson's correlation ignored scale differences between empirical and simulated static FC, which led to pMFM-simulated static FC values being systematically larger than empirical FC values (Fig. 2C). In future studies, we will explore an additional cost term that penalizes absolute differences between empirical and static FC.

The disagreement between the simulated and empirical FCD matrices was defined as the Kolmogorov–Smirnov (KS) distance between the probability distribution functions (pdfs) constructed from the upper triangular entries of the two matrices[22]. The pdf of an FCD matrix was constructed by collapsing the upper triangular entries of the matrix into a histogram and normalized to have an area of one. A smaller KS distance indicated a more similar FCD. To optimize fit to both static FC and FCD, an overall cost was defined as $(1 - r) + KS$. Thus lower cost implies a better fit to static FC and FCD.

To minimize the cost function in the training set, we seek to compute an "average" FCD matrix. We note that FCD matrices could not be directly averaged across rs-fMRI runs and participants because there was no temporal correspondence across runs during the resting-state. Because the goal here was to compute the KS distance, we simply averaged the pdfs from the FCD matrices all the runs of all participants within the training set, which we referred to as average FCD pdf. When evaluating KS distance in the validation and test sets, average FCD pdfs were also computed using the same approach.

**Optimization procedure**. To optimize the cost function, we considered three algorithms: covariance matrix adaptation evolution strategy (CMA-ES)[38], self-organizing migrating algorithm (SOMA[59]) and hyperparameter optimization using radial basis functions and dynamic coordinate search (HORD[60]).

Given a particular random initialization of the 10 unknown parameters, the three algorithms (CMA-ES, SOMA, HORD) were applied to the HCP training set. Each algorithm was iterated 500 times, generating 500 candidate parameter sets. This procedure was repeated 10 times, yielding 5000 candidate parameter sets. For each algorithm, the 5000 candidate parameter sets were evaluated in the validation set to obtain the top 10 candidate parameter sets. To ensure diversity among the parameter sets, the procedure to select the top 10 parameter sets was as follows. First, the parameter set with the lowest validation cost was selected. Then, the parameter set with the lowest validation cost and whose parameter maps exhibited less than 0.98 correlation with the current selected parameter set(s) was selected. This procedure was repeated until 10 parameter sets were selected. Across the three algorithms, CMA-ES performed the best in the validation set (Supplementary Fig. S1), so this study focused on CMA-ES.

The top 10 candidate parameter sets from CMA-ES were then applied to the HCP test set SC. For each parameter set, 1000 simulations were performed, yielding 1000 simulated static FC and FCD matrices. The 1000 simulated FC and FCD pdfs were then averaged, yielding an average simulated FC and an average simulated FCD pdf. Pearson's correlation was then computed between the average simulated FC and the average empirical FC from the HCP test set. Similarly, KS statistics was computed between the average simulated FCD pdf and the average empirical FCD pdf from the HCP test set.

We note that by collapsing the entries of the FCD matrix into a pdf, we were ignoring the recurrent structure in the FCD matrix. Supplementary Fig. S17 shows the FCD pdfs of empirical and simulated data. At the individual-level, the FCD pdf exhibited a bimodal distribution. Because the FCD pdfs were shifted across participants, the group-level FCD pdf was unimodal. Although the pMFM was fitted to the group-level FCD pdf, the resulting FCD distribution exhibited hints of bimodality and recurrent structure similar to empirical FCD (Fig. 2).

**Statistical test of correlation between first derivatives of FCD mean and SW-STD**. To quantify the correspondence between FCD mean and SW-STD (Fig. 5), the correlation between the first derivative of the FCD mean time course and the first derivative of the SW-STD time course was computed for each cortical region (Fig. 6). To compute the statistical significance of the correlations, fMRI runs were permuted across participants. For each ROI, the FCD-STD correlations were recomputed and averaged across runs and participants, yielding a single null correlation value. This permutation procedure was repeated 10,000 times, so that a null distribution of correlations was obtained for each ROI.

**Causal perturbations of pMFM**. To more directly link sensory-motor regions with FCD, we tested whether perturbation of sensory-motor regions can "kick" the system from an incoherent FCD state to a coherent FCD state. Among 1000 random simulations of the pMFM, time segments in the incoherent (low FCD mean) state lasting for at least 200 contiguous fMRI time points (TRs) were selected, yielding 300 time segments. Low FCD mean was defined as being <0.6.

Perturbation was applied to the neural signals (synaptic gating variable $S_i$) of the top 5 regions whose SW-STD correlated with FCD (Fig. 6B). We note that during the incoherent state, the values of the synaptic gating variables could be low or high. To increase the amplitude of the neural signals, we would decrease (or increase) the synaptic gating variables if they were high (or low). More specifically, let $S_{max}$ and $S_{min}$ be the maximum and minimum synaptic gating variable values across all cortical regions. When the neural signal was low, we set $S_{t+\delta t} = S_t + 0.8(S_{max} - S_t)$, where $\delta t$ corresponded to the resolution of the simulations, which is 0.01 s in the current study. When the neural signal was high, we set $S_{t+\delta t} = S_t - 0.8(S_t - S_{min})$. The perturbations were applied for 72 iterations, corresponding to 1 TR in the simulated fMRI signal.

**Gene expression analysis**. Publicly available human gene expression data from six postmortem donors (1 female), aged 24–57 years (42.5 ± 13.4) were obtained from the Allen Institute[61]. Processing followed the pipeline from Anderson and colleagues[30] (https://github.com/HolmesLab/2020_NatComm_interneurons_cortical_function_schizophrenia), yielding 17,448 brain-expressed genes and 1683 analyzable cortical samples. Our analyses in turn focused on 2413 brain-specific genes[26,62]. Z-normalized gene expression values of parvalbumin (PVALB) and somatostatin (SST) were

averaged within each cortical region and the difference was computed. The FCD-STD correlation maps (Fig. 6) were correlated with the PVALB-SST spatial map (Fig. 7).

To establish statistical significance, we considered two approaches. First, we considered the spin test. The parcellations were randomly rotated. For each rotated parcellation, we recomputed the PVALB-SST difference and correlated the resulting gene expression maps with the FCD-STD correlation maps, yielding a single null correlation value. This was repeated 1000 times yielding a complete null distribution.

To test the specificity of PVALB-SST, we performed random-gene-pair tests. A random pair of genes were selected from the 2413 brain-specific genes[26]. Gene expression difference between the random gene pairs was computed and correlated with the STD-FCD correlation maps generating a null correlation value. This was repeated 10,000 times yielding a complete null distribution.

**Reporting summary**. Further information on research design is available in the Nature Research Reporting Summary linked to this article.

## Data availability
The group-level FC, SC, and FCD cumulative distribution functions used in this study have been deposited in the Zenodo[63] database under accession code 5518257. The raw diffusion MRI, rs-fMRI, and T1w/T2w data are publicly available (https://www.humanconnectome.org/study/hcp-young-adult/document/1200-subjects-data-release). Source data are provided with this paper.

## Code availability
The code used in this paper is deposited in Zenodo[63] database under accession code 5518257. The code was reviewed by one co-author (S.Z.) to reduce the chance of coding errors. The software dependencies are MATLAB (2018b); Python (3.6); Pytorch (1.0.1). From time to time, the code might be updated. The most updated version of the code can be found on GitHub (https://github.com/ThomasYeoLab/CBIG/tree/master/stable_projects/fMRI_dynamics/Kong2021_pMFM).

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

## Acknowledgements

This work was supported by the Singapore National Research Foundation (NRF) Fellowship (Class of 2017), the NUS Yong Loo Lin School of Medicine (NUHSRO/2020/124/TMR/LOA), the Singapore National Medical Research Council (NMRC) LCG (OFLCG19May-0035), and the United States National Institutes of Health (R01MH120080). Any opinions, findings, and conclusions, or recommendations expressed in this material are those of the authors and do not reflect the views of the Singapore NRF or the Singapore NMRC. Our computational work was partially performed on resources of the National Supercomputing Centre, Singapore (https://www.nscc.sg). Data were in part provided by the Human Connectome Project, WU-Minn Consortium (Principal Investigators: David Van Essen and Kamil Ugurbil; 1U54MH091657) funded by the 16 NIH Institutes and Centers that support the NIH Blueprint for Neuroscience Research; and by the McDonnell Center for Systems Neuroscience at Washington University.

## Author contributions

X.K., R.K., P.W., C.O., K.A., A.H., J.D.M., G.D., M.v.d.H. and B.T.T.Y. designed the research. X.K. conducted the research. X.K., R.K., C.O. and B.T.T.Y. analyzed and interpreted the results. X.K. and B.T.T.Y. wrote the manuscript and made figures. X.K. analyzed the data. X.K. and S.Z. reviewed and published the code. All authors provided analytic support. All authors edited the manuscripts.

## Competing interests

The authors declare no competing interests.
