## [Peer Review File · Nature Communications]

REVIEWER COMMENTS

Reviewer #1 (Remarks to the Author):

Kong et al explore the contribution of slowly varying gradients of functional and morphological cortical properties to whole-brain dynamics. They combine analyses of Human Connectome data with computational modelling, showing that indeed incorporating these properties leads to a better match between predicted and observed dynamics.

The paper provides additional insights into the nature of whole brain dynamics, an area of current interest in systems neuroscience. For example, showing that the parameter of external noise decreases from sensorimotor to association cortex whereas recurrent connection strength and noise increase is a very interesting finding with novel and strong face validity (given one expects external input to be greatest in sensory cortex). Likewise, the perturbation analyses of the simulated system (showing that sensorimotor perturbations could knock the system between modes) is interesting and novel.

I think this is a strong candidate for publication in Nature Communications. I do have a number of technical concerns that I think the authors could address through a major revision. I have also suggested some additional analyses which the authors could pursue to improve insights into the dynamics.

1. As a very crude first order principle, isn't there an element of circularity in using the first functional connectivity gradient to better fit modelled functional connectivity, in the sense that one is incorporating information derived from functional connectivity to find models that better fit functional connectivity? I don't think out of sample modelling addresses this, as the highly reliable presence of functional gradients across people has previously been well established [1], and additionally because the simpler models have already been filtered out before the cross-validation by a procedure with no penalty for complexity. And btw I couldn't see summary statistics of the difference between performance on the training and test data sets.

2. How uniquely identifiable are each of the ten free parameters from the fitting procedure – some evidence should be added to show they are well behaved (e.g. limited number of maxima + sufficiently statistically independent to allow sufficient disambiguation). Even though the FCD are high dimensional, it is not clear how much dimension reduction on these matrices was performed before fitting the models (see point 3 below).

3. What property of the empirical and simulated FCD are responsible for a good fit (small KS stat)? Clearly not the exact sequence of states otherwise it seems highly unlikely to cross-validate? On reading, it seems likely the overall histogram of time-lagged correlation coefficients? What do these

look like and what aspect of these do the pMFM capture? I am guessing the empirical histograms are rather bimodal, with a high prevalence of high values near the short-time delays and the highly correlated off-diagonal blocks; A second mode of near zero CC's seems likely.

If the fitting of the FCD's also incorporate the time delay, then the whole recurrence structure of the dynamics are captured – quite interesting. If this is the case, the authors could consider looking for nonlinear structure by phase randomizing the time series and comparing the recurrence plots recovered from these null data – see Fig. 5 of [2].

4. The correlation of the empirical and fitted z-trans FC (Fig2C) is not particularly linear, with a crowding above the line of best fit – please comment.

5. “When recurrent connectional strength w , external input current I , and noise amplitude σ were optimized by CMA-ES, but constrained to be spatially homogeneous (Figure 3), then there was substantially weaker agreement with empirical static FC ($r = 0.56 \pm 0.05$) and FCD (KS = 0.50 ± 0.30).” – adding an homogenous constraint decreases the degrees of freedom and can ONLY decrease the fit –not sure what this weakest fit brings to our understanding. The same holds for only using T1w/T2w and even though the authors concede this point, again not sure what this adds without an appropriate penalty to compensate for the increase in effective degrees of freedom.

6. The transition between the two states of low and high global coherence is very similar to the transitions between low and high “efficiency” in Figure 3 of [3] – since the data are sampled from the same project, these likely ARE the same phenomena – I think this should be cited. Since this emerged in [3] by simulated a uniform NMM on the connectivity matrix, it seems unlikely that the gradient-informed local parameters are required for this.

More substantially, deeper insights into the dynamics could be obtained by looking at the relationship of the amplitude and variance in these two ‘modes’, and also from examining the lifetime distributions – both as per Freyer et al (2012), already cited. Ideally this would encompass the entire distribution of states (/correlations) and can be used to disambiguate meta- from multi-stability. As it stands, partitioning the data into top and bottom deciles and performing statistics on these seems quite circular.

Minor:

Lines 374-5 – these should all refer to Figure 6, not 7

References:

1. Margulies, et al (2016). Situating the default-mode network along a principal gradient of macroscale cortical organization. *Proceedings of the National Academy of Sciences*, 113(44), 12574-12579.
2. Roberts et al (2019). Metastable brain waves. *Nature communications*, 10(1), 1-17.
3. Zalesky, et al (2014). Time-resolved resting-state brain networks. *Proceedings of the National Academy of Sciences*, 111(28), 10341-10346.

Reviewer #2 (Remarks to the Author):

Anatomical and Functional Gradients Shape Dynamic Functional Connectivity in the Human Brain

Xiaolu Kong, Ru Kong, Csaba Orban, Wang Peng, Shaoshi Zhang, Kevin Anderson, Avram Holmes, John D. Murray, Gustavo Deco, Martijn van den Heuvel, B.T. Thomas Yeo

Summary

This study simulates static and dynamic functional connectivity using group-level information about functional gradient and myelination to inform heterogeneity across the cortex. The simulated results show improved correspondence to empirical data compared with a model that assumed homogeneity. The findings furthermore suggest the presence of high and low coherence states, which were causally linked to signal amplitude in sensory-motor regions in follow-up analyses. Associations between spatial amplitude patterns and gene expression were also shown.

General

This is a well-written and very thorough study that provides important new insights into neural and genetic mechanisms that shape patterns of functional connectivity. The control analyses are appropriate and preemptively addressed most of the questions that came to my mind while reading the article. I just have a small number of relatively minor questions, see below.

Major comments

1. The functional gradient maps essentially reflect the principal direction of spatial organization in static functional connectivity. These maps are not independent from the static FC estimated in this paper and there is a potential concern of circularity in their use to simulate static FC. This is consistent with the results in Fig 3C showing good performance in predicting static FC using the gradient maps alone. This is not a major issue and it is actually of interest to bridge these different representations of static FC (i.e., gradients vs parcellated FC). Nevertheless, this should be clarified and discussed in the manuscript to help the reader in interpreting the results.

2. Weaker agreement is shown between the proposed model and several other tested models (assumed spatial homogeneity/ gradient only/ T1-T2 only etc; Figs 2&3). However, no statistical testing is performed for some of these comparisons. I would encourage the authors to include statistical comparisons.

Minor comments

A. Group-level gradient and T1/T2 maps are used to inform the model. This is appropriate for this first study, which is already comprehensive in many other respects. But I wonder if the use of group-level maps limits the degree to which simulated data are able to capture between-subject variability present in empirical data. This is an interesting area for future research and I would be interested in the author's thoughts on this front. Perhaps it is something to consider including in the discussion.

B. The first functional gradient is used to inform spatial heterogeneity. I would be interested in how using additional gradients may influence the results, especially in relation to the amplitude-state associations. Again, I appreciate that this might be outside of the scope of the current paper and more a direction for future research.

Janine Bijsterbosch

Reviewer #3 (Remarks to the Author):

This manuscript describes a new computational modeling approach which seeks to estimate observed patterns of static and dynamic cortical functional connectivity (FC and FCD respectively) from group-level estimates of structural connectivity, and region-specific cortical myelination (T1w/T2w ratio) and a primary FC spatial gradient. The major advance of this model on prior work is that it models observed FC and FCD using a spatially heterogeneous cortical mean field model where key local circuit properties are parameterized using the local group-average measures of myelination and primary FC gradient. The authors show that use of region-specific information improved model

fit. They then use the winning models to test hypotheses re the role of amplitude variability in rsfMRI signals in “flipping” between periods of high and low-coherence in FCD - concluding that sensorimotor regions may drive this behavior. Finally, the authors show that regional variation in rsfMRI signal amplitude from their computational model is correlated with regional variation in the difference in molecular markers of parvalbumin and somatostatin interneurons from the Allen Human Brain Atlas.

I think this study will be of interest to the research community, and is notable for several strengths: the authors pursue models that are more realistic than prior ones, by virtue of allowing local circuit features to vary across the cortical sheet; there is a rigorous approach to model selection with extensive sensitivity analyses to test the influence of important methodological decisions; the linkage to gene-expression data is also of value, and the authors do this with a sensible discussion of caveats and limitations.

The authors of this work are leaders in the generation and implementation of biophysical models for the brain, so I would anticipate that this aspect of the work is sound. However, I do not have expertise in biophysical modeling of brain activity, and it would be important that this expertise is represented within the peer-review process.

I think the following issues may benefit from further consideration:

1. The T1w/T2w and FC gradient are just two of many cortical features that all capture the same “primary gradient” spanning sensorimotor and cortical. Others would be structural/functional variability across people, first PC of expression or sets of genes loading highly onto it etc. Is there a way of running “control” versions of the model that swap out T1w/T2w and/or FC gradient for these alternatives - to show that these two features perform best. These two features are sensible to consider given the model features of recurrent connections, external input and noise. But the set-up in this paper provides a nice opportunity to pit facets of the primary gradient against each other/

2. The T1w/T2w and FC gradient seem to be making differential contributions for static FC (where FC does most of the “work”) vs. FCD (where T1/T2 seems to get you pretty close to the combined model). Would be good to see some more analytic exploration and/or Discussion of this.

3. I correlation matrix for inter-regional variation in T1w/T2w, FC gradient, w , I and σ would be good to include (as sup maybe) -perhaps also alongside some of the other imaging and expression-derived features mentioned in comment #1 above. My general point is that many of these results seem to hit on the same primary gradient, and I think this means we really need to see just how different the input maps, parameter maps (Fig 4) and others (slide 6, 7) are.

4. Inter-individual variation: This may reflect a limitation in my understanding of the method, but ... it seemed to me that the first section on predicting FC and FCD from structural connectivity, T1w/T2w and FC gradient through the dynamic mean field model could be done at the individual level too? If this is correct, then that would provide a good way to test the stability of each feature's utility to the model across people, and provide an added level of association.

We are pleased to see a high level of enthusiasm for our work. We thank the reviewers for their close read of this manuscript and insightful comments. Several important suggestions were made for improvement. We have considered each suggestion carefully and revised accordingly. Please find detailed responses (in blue) to the reviewer comments (in italics). For convenience, changes to the manuscript are quoted verbatim (normal font) when appropriate. We believe the manuscript is much improved and hope it is now suitable for publication.

We note that the title of the study has now been changed to “Sensory-Motor Cortices Shape Functional Connectivity Dynamics in the Human Brain” in line with what we believe is the most robust and interesting result in this study as described in the revised manuscript.

Reviewer #1:

(RIQ1) Kong et al explore the contribution of slowly varying gradients of functional and morphological cortical properties to whole-brain dynamics. The combine analyses of Human Connectome data with computational modelling, showing that indeed incorporating these properties leads to a better match between predicted and observed dynamics.

The paper provides additional insights into the nature of whole brain dynamics, an area of current interest in systems neuroscience. For example, showing that the parameter of external noise decreases from sensorimotor to association cortex whereas recurrent connection strength and noise increase is a very interesting finding with novel and strong face validity (given one expects external input to be greatest in sensory cortex). Likewise, the perturbation analyses of the simulated system (showing that sensorimotor perturbations could knock the system between modes) is interesting and novel. I think this is a strong candidate for publication in Nature Communications. I do have a number of technical concerns that I think the authors could address through a major revision. I have also suggested some additional analyses which the authors could pursue to improve insights into the dynamics.

We thank the reviewer for the positive comments.

(RIQ2) As a very crude first order principle, isn't there an element of circularity in using the first functional connectivity gradient to better fit modelled functional connectivity, in the sense that one is incorporating information derived from functional connectivity to find models that better fit functional connectivity? I don't think out of sample modelling addresses this, as the highly reliable presence of functional gradients across people has previously been well established [1], and

additionally because the simpler models have already been filtered out before the cross-validation by a procedure with no penalty for complexity. And btw I couldn't see summary statistics of the difference between performance on the training and test data sets.

The reviewer is concerned about potential circularity in using the first functional connectivity (FC) gradient for the parametric mean field model (pMFM). We share the reviewer's concern on the importance of avoiding circularity in analyses, however, we do not believe it is an issue with our present approach.

1) We note that the FC gradient was estimated from static FC, so we hope the reviewer agrees that there is no circularity when it comes to pMFM predicting functional connectivity dynamics (FCD) well.

2) Furthermore, given that the FC gradient was derived from the training set, we do not believe the analysis is circular even in the case of static FC. For example, many studies utilized parcellations (e.g., Gordon, Schaefer, Shen, etc) derived from resting-state FC to study resting-state FC in new participants. As far as we know, nobody thinks that it is circular to use resting-state parcellations for resting-state analyses. To draw a more concrete parallel with our current study, we note that the Schaefer parcellation was derived from the resting-state FC of 744 participants. In one analysis (Schaefer et al., 2018), we showed that vertices within Schaefer parcels exhibited more similar resting-state fMRI time courses than other parcellations in a second set of 745 participants (Schaefer et al., 2018). There is no circularity in this analysis because the parcellations were estimated and tested on two non-overlapping groups of participants. Similarly, in the current study, the MFM parameters were estimated and then tested on non-overlapping groups of participants. Therefore, while it might seem obvious that parameterizing the MFM using resting-FC would help to explain out-of-sample resting-FC better, it is not circular.

3) The reviewer commented that “the simpler models have already been filtered out before the cross-validation by a procedure with no penalty for complexity”. We apologize for being unclear, but this is not the case in our study. To elaborate, the figure below corresponds to Figure 1B in the manuscript. The HCP training, validation and test sets comprised non-overlapping participants. The HCP training set was used to generate 5000 MFM parameter sets, which were then evaluated in the HCP validation set. The top 10 candidates from the validation set were then evaluated in the HCP test set. We note that the exact same procedure was applied to different parameterizations of the MFM: (1) T1/T2 + RSFC gradient, (2) T1/T2 only, (3) RSFC gradient only and (4) spatially constant parameters. Thus, the quantitative comparison of the four different parameterizations using the HCP test set (Figure 3 in manuscript) is valid and non-circular.

4) The reviewer suggests that the cross-validation framework needs to penalize complexity. We would like to emphasize that evaluation with an independent test set obviates the need to penalize for complexity. In the toy example below, the 10-degree polynomial (blue curve) can fit the 11 black data points exactly. However, the blue curve cannot predict a new data point (red star) well. On the other hand, the linear black line does not fit the 11 black data points exactly but is able to predict the red star better. Therefore, the blue curve (with more parameters) has better fit in the training set (black dots), but worse fit in the test set (red star). In other words, evaluation in the test set reveals the black curve to be the better model despite it having less parameters and worse fit in the training set.

5) As requested by the reviewer, we now report summary statistics of the top 10 candidates (from the validation set) in the training, validation and test sets in Table S1 (see below). The pMFM was statistically better than the spatially homogeneous MFM for both metrics (FC correlation and FCD KS statistic) in the test set. Compared with other alternative parameterizations, pMFM was statistically better in one metric and statistically comparable in the other metric.

6) It is also important to point out the example that if the recurrent connectional strength w , external input current I , and noise amplitude σ were allowed to be spatially heterogeneous across brain regions, but not constrained by T1w/T2w or FC gradient (i.e., non-parametric), then simulations could achieve realistic static FC, but not FCD in test set. This is an example, where a more flexible model (205 free parameters) yielded excellent fit in the training and validation sets but significantly worse fit in the test set (see “non-parametric” row in Table S1).

We thank the reviewer for the comments, which suggests the insufficient clarity in our writing and the need to clarify the above points in the manuscript. We have updated the manuscript as follows.

pg 7 (Figure 1 captions)

Comparison of the pMFM with other parametrizations utilized the same training-validation-test procedure.

pg 8 (Results)

Both FC gradient and T1w/T2w map were estimated from the training set.

pg 8 (Results)

The pMFM was compared with other parametrizations using the same training-validation-test procedure.

pg 9 (Results)

Across the 10 best candidate sets from the validation set, KS distance between empirical and simulated FCD was 0.12 ± 0.03 (mean \pm std) in the test set.

pg 10 (Results)

Across the 10 best candidate sets from the validation set, correlation between empirical and simulated static FC was 0.66 ± 0.03 in the test set.

pg 10 (Results)

When recurrent connectional strength w , external input current I , and noise amplitude σ were optimized by CMA-ES, but constrained to be spatially homogeneous (Figure 3), there was substantially weaker agreement with empirical static FC ($r = 0.55 \pm 0.05$) and FCD ($KS = 0.50 \pm 0.31$) in the test set.

pg 11 (Results)

Table S1 provides summary statistics of the two metrics (FC correlation and FCD KS statistic) in the training, validation, and test sets. The pMFM was statistically better than the spatially homogeneous MFM for both metrics in the test set. Compared with other alternative parameterizations, pMFM was statistically better in one metric and statistically comparable in the other metric.

pg 12 (Figure 3 captions)

Agreement (Pearson's correlation) between simulated and empirically observed static FC, as well as disagreement (KS distance) between simulated and empirically observed FCD across different conditions in the test set ... * indicates statistical significance after correcting for multiple comparisons with a false discovery rate of $q < 0.05$. All p values are reported in Table S1.

pg 13 (Results)

It is worth emphasizing that the different parameterizations were compared with the same training-validation-testing procedure (Figure 1B), which automatically controls for model complexity or degrees of freedom. A more complex model will generally fit the training data better but might not perform well in the test set. For example, when recurrent connectional strength w , external input current I , and noise amplitude σ were allowed to be spatially heterogeneous across brain regions, but not constrained by T1w/T2w or FC gradient (i.e., non-parametric), then simulations could achieve realistic static FC, but not FCD in test set (Table S1). This is an example, where a more flexible model (205 free parameters) yielded excellent fit in the training and validation sets but significantly worse fit in the test set.

pg 13 (Results)

On the other hand, FC gradient was better than T1w/T2w map at explaining static FC across the three experiments (Figures 3 and S4), which made intuitive sense given that FC gradient was derived from static FC. However, we note that the analyses were not circular given that the FC gradient was derived from the training set and performance was evaluated on the test set.

pg. 29 (Discussion)

Furthermore, while it made intuitive sense that utilizing the resting-state FC gradient would help to explain resting-fMRI dynamics, the training-validation-test scheme ensures the analysis is not circular.

	FC Correlation Training	FC Correlation Validation	FC Correlation Test	FC p-value Test Set	FCD KS Statistics Training	FCD KS Statistics Validation	FCD KS Statistics Test	FCD p- value Test
pMFM	0.66±0.03	0.67±0.03	0.66±0.03		0.04±0.02	0.03±0.01	0.12±0.03	
Gradient only	0.63±0.06	0.63±0.06	0.63±0.06	0.188	0.15±0.15	0.20±0.18	0.31±0.26	0.031
T1w/T2w only	0.56±0.04	0.56±0.04	0.55±0.03	2.0e-5	0.07±0.05	0.09±0.05	0.15±0.06	0.123
Constant w, I & σ	0.58±0.03	0.55±0.04	0.55±0.05	2.4e-5	0.28±0.30	0.31±0.29	0.50±0.31	0.001
Constant w	0.63±0.03	0.64±0.03	0.62±0.03	0.0245	0.05±0.02	0.07±0.04	0.11±0.04	0.729
Constant I	0.58±0.06	0.59±0.07	0.58±0.06	0.0037	0.11±0.06	0.13±0.08	0.14±0.06	0.179
Constant σ	0.57±0.03	0.59±0.04	0.57±0.04	2.48e-5	0.09±0.03	0.08±0.03	0.24±0.21	0.088
Non- parametric	0.66±0.02	0.66±0.03	0.66±0.03	0.754	0.06±0.03	0.04±0.02	0.28±0.11	2.1e-4
FC Cost only	0.74±0.005	0.76±0.002	0.73±0.01	2.1e-6	0.87±0.005	0.87±0.005	0.88±0.01	7.1e-23

Table S1. Model performance in the HCP training, validation, and test sets. The HCP training set was used to generate 5000 MFM parameter sets, which were then evaluated in the HCP validation set. The top 10 candidates from the validation set were then evaluated in the HCP test set. The table below shows the performance (mean \pm std) of these top 10 candidates in the HCP training, validation and test sets. Higher FC correlations and lower FCD KS statistics indicate better fit. The last column showed the p values of two-sample two-tail t-tests between pMFM and other parameterizations or cost function in the test set. Bolded results indicate statistically significant results after false discovery rate (FDR) correction ($q < 0.05$). Our approach (pMFM) was statistically better than (or comparable to) all other parameterizations in terms of FC correlation and/or FCD KS statistic in the test set. Optimizing for only fit to static FC (last row in table) led to slightly better (but statistically significant) fit to static FC in the test set, but much worse (and statistically significant) fit to FCD in the test set.

(RIQ3) How uniquely identifiable are each of the ten free parameters from the fitting procedure – some evidence should be added to show they are well behaved (e.g. limited number of maxima + sufficiently statistically independent to allow sufficient disambiguation). Even though the FCD are high dimensional, it is not clear how much dimension reduction on these matrices was performed before fitting the models (see point 3 below).

The reviewer has raised a very good point. We do not think that the ten free parameters are identifiable since the T1w/T2w map and FC gradient are strongly correlated, so different linear combinations of the two maps can lead to similar local circuit parameter maps.

However, we do not believe this is an issue because ultimately, we are interested in the local circuit parameter maps rather than the coefficients of the linear model. Therefore, we believe it is more important that the local circuit parameter maps be somewhat identifiable. Given the MFM is a highly nonlinear model, we expect the maps to not be fully identifiable. Therefore, the question is not a yes/no question but rather the degree of the identifiability. This is a question we have explored in the original manuscript.

More specifically, in the original manuscript, we explored the top ten parameter sets from the validation set and found good correspondences across the ten parameter sets for external input I and noise amplitude σ . In the case of recurrent connection strength w , we found that one of the ten parameter sets exhibited the opposite gradient direction. The following text was in the original (and current) manuscript, replicated here for your convenience.

In the case of recurrent connection strength, one of the ten parameter sets exhibited the opposite direction (i.e., decrease from sensory-motor regions to association networks; Figure S5), suggesting potential degeneracy in the case of recurrent connection strength.

Given this question from the reviewer, we decided to explore this issue in more detail. In particular, the original analysis was “biased” to find degeneracy because our algorithm tried to avoid parameter estimates that were too similar. Therefore, we performed another analysis in which we compared all 5000 candidate parameter sets from the training set with the top parameter set from the validation set. In general, parameter sets with very good validation cost were strongly correlated with the top parameter set from the validation set. The manuscript has been updated as follows.

pg. 14 (Results)

The previous analysis was “biased” to find degeneracy given that the top 10 parameter sets were selected to ensure diversity (see Methods). To further explore the degeneracy issue, the recurrent connection strength map of the top parameter set (from the validation set) was correlated with the recurrent connection strength maps of the remaining 4999 candidate parameter sets (Figure S6). In general, parameter sets with good validation cost were strongly correlated with the top parameter estimate from the validation set. Similar conclusions were obtained for external input and noise amplitude, although external input appeared to be less stable than recurrent connection strength and noise amplitude.

Note that the original manuscript forgot to mention that the top 10 parameter sets were selected to ensure some level of diversity. We have updated the manuscript with this additional information.

pg 35 (Methods)

To ensure diversity among the parameter sets, the procedure to select the top 10 parameter sets was as follows. First, the parameter set with the lowest validation cost was selected. Then, the parameter set with the lowest validation cost and whose parameter maps exhibited less than 0.98 correlation with the current selected parameter set(s) was selected. This procedure was repeated until 10 parameter sets were selected.

Figure S6. Correlations of local circuit parameter maps between the top parameter set from the validation set and the remaining 4999 candidate parameter sets. The 4999 candidate parameter sets were ordered based on their performance in the validation set. In general, parameter sets with good validation cost were strongly correlated with the top parameter estimate from the validation set. In the case of recurrent connection w , the correlations between the top 400 estimates of w and the top estimate of w were more than 0.9. In the case of external input I , the correlations between the top 60 estimates of I and the top estimate of I were more than 0.9. In the case of noise amplitude σ , the correlation between the top 400 estimates of σ and the top estimate of σ were more than 0.9. In the case of external input I . Therefore, external input appeared to be less stable than recurrent connection strength and noise amplitude.

(R1Q4) What property of the empirical and simulated FCD are responsible for a good fit (small KS stat)? Clearly not the exact sequence of states otherwise it seems highly unlikely to cross-validate? On reading, it seems likely the overall histogram of time-lagged correlation coefficients? What do these look like and what aspect of these do the pMFM capture? I am guessing the empirical histograms are rather bimodal, with a high prevalence of high values near the short-time delays and the highly correlated off-diagonal blocks; A second mode of near zero CC's seems likely.

If the fitting of the FCD's also incorporate the time delay, then the whole recurrence structure of the dynamics are captured – quite interesting. If this is the case, the authors could consider looking for nonlinear structure by phase randomizing the time series and comparing the recurrence plots recovered from these null data – see Fig. 5 of [2].

We apologize for not being sufficiently clear. Indeed, we only consider the overall histogram of the FCD matrix, so the recurrent structure and state sequence are ignored. In other words, the probability distribution function (pdf) of an FCD matrix is constructed by collapsing the upper triangular entries of the matrix into a histogram and normalized to have an area of one.

At the individual-level, the pdf of the FCD matrix does exhibit a bimodal distribution (see figure below). Because the FCD pdfs are shifted across participants, the group-level FCD pdf is unimodal. Interestingly, even though the pMFM was fitted to the group-level pdf, the resulting FCD distribution exhibits hints of bimodality and the recurrent structure.

Another interesting point is that even though the recurrent structure in the FCD matrix was ignored, the dwell times of the two states were similar in empirical and simulated data (see our response to R1Q6).

We have updated the manuscript as follows.

pg 34 (Methods)

The pdf of an FCD matrix was constructed by collapsing the upper triangular entries of the matrix into a histogram and normalized to have an area of one.

pg 36 (Methods)

We note that by collapsing the entries of the FCD matrix into a pdf, we were ignoring the recurrent structure in the FCD matrix. Figure S17 shows the FCD pdfs of empirical and simulated data. At the individual-level, the FCD pdf exhibited a bimodal distribution. Because the FCD pdfs were shifted across participants, the group-level FCD pdf was unimodal. Although the pMFM was fitted to the group-level FCD pdf, the resulting FCD distribution exhibited

hints of bimodality and recurrent structure similar to empirical FCD (Figure 2).

Figure S17. Functional connectivity dynamics (FCD) probability distribution functions (pdfs) of empirical and simulated data. At the individual-level, the FCD pdf was bimodal. At the group-level, the FCD pdf was unimodal. Interestingly, although the pMFM was fitted to the group-level FCD pdf, the resulting FCD distribution still exhibits hints of bimodality.

(R1Q5) The correlation of the empirical and fitted z-trans FC (Fig2C) is not particularly linear, with a crowding above the line of best fit – please comment.

The reviewer raised a very good point. Despite the strong correlation between empirical and simulated static FC, the simulated FC values were generally higher than the empirical FC values. This scale difference arose because we utilized “correlation” in our optimization criterion, which ignored scale differences between the empirical and simulated FC. In future studies, we will explore an additional cost term penalizing absolute differences between empirical and static FC. This point is now further elaborated in the manuscript

pg 34 (Methods)

The agreement between the simulated and empirical static FC matrices was defined as the Pearson’s correlation (r) between the z-transformed upper triangular entries of the two matrices. Larger r indicated more similar static FC. Pearson’s correlation was chosen given its popularity in the literature. However, we note that Pearson’s correlation ignored scale differences between empirical and simulated static FC, which led to pMFM-simulated static FC values being systematically larger than empirical FC values (Figure 2C). In

future studies, we will explore an additional cost term that penalizes absolute differences between empirical and static FC.

(R1Q6) “When recurrent connectional strength w , external input current I , and noise amplitude σ were optimized by CMA-ES, but constrained to be spatially homogeneous (Figure 3), then there was substantially weaker agreement with empirical static FC ($r = 0.56 \pm 214 0.05$) and FCD ($KS = 0.50 \pm 0.30$).” – adding an homogenous constraint decreases the degrees of freedom and can ONLY decrease the fit –not sure what this weakest fit brings to our understanding. The same holds for only using T1w/T2w and even though the authors concede this point, again not sure what this adds without an appropriate penalty to compensate for the increase in effective degrees of freedom.

As stated in our previous response to R1Q2, our perspective is different from the reviewer. Just to reiterate, having more degrees of freedom will improve the quality of fit in the training set, but there is no guarantee that the fit will improve in the test set. Please see further elaboration under points 4 and 6 in our response to R1Q2. Updates to the manuscript are also found in our response to R1Q2.

Even though we do not believe it is necessary to control for model complexity, to address the reviewer’s comment, we performed a control experiment in which the local circuit properties of pMFM were parameterized with randomly rotated versions of T1w/T2w map and/or FC gradient. Due to limited computational resources, we performed 1000 random rotations and picked the rotated map with the smallest absolute correlation with the original T1w/T2w map (or FC gradient). The MFMs with rotated parameterizations were optimized using the same training-validation-test procedure in Figure 1B. Despite having the same degrees of freedom as the original pMFM, the rotated parameterizations led to worse fit to static FC and/or FCD in the test set (see below).

The manuscript has been updated as follows.

pg 24 (Results)

We have shown that combining T1w/T2w map and FC gradient led to more realistic brain dynamics than using either no gradient or only one gradient (Figure 3). To further explore the specificity of the parameterization, we repeated the training-validation-test procedure (Figure 1B) using randomly rotated versions of T1w/T2w map and/or FC gradient. Despite having the same degrees of freedom as the original pMFM, the rotated parameterizations led to worse fit to static FC and/or FCD in the test set (Figure S9).

Figure S9. Parameterization with FC gradient and T1w/T2w map led to more realistic static FC and FCD compared with randomly rotated versions of T1w/T2w map and/or FC gradient in the test set. To generate the rotated versions of T1w/T2w map (or FC gradient), we performed 1000 random rotations and picked the rotated map with the smallest absolute correlation with the original T1w/T2w map (or FC gradient). The MFMs with rotated parameterizations were optimized using the same training-validation-test procedure in Figure 1B. * indicates statistical significance after multiple comparisons correction with false discovery rate (FDR) $q < 0.05$.

(R1Q7) The transition between the two states of low and high global coherence is very similar to the transitions between low and high “efficiency” in Figure 3 of [3] – since the data are sampled from the same project, these likely ARE the same phenomena – I think this should be cited. Since this emerged in [3] by simulated a uniform NMM on the connectivity matrix, it seems unlikely that the gradient-informed local parameters are required for this.

The reviewer has raised a good point. Indeed, the FCD matrix (inspired by Hansen et al., 2015) is only one approach to capture time-varying FC. The sliding-window network efficiency by Zalesky and colleagues is equally valid (Zalesky et al., 2014) and we can probably come up with a cost function to optimize for sliding-window network efficiency. Indeed, the reviewer is correct that the high and low efficiency states do correspond to the high and low coherent states in the FCD matrix (see figure below).

The reviewer is also correct that Zalesky and colleagues were able to generate high and low efficiency states with a spatially uniform MFM. However, we note that they did not utilize a training-testing scheme to explore generalizability. Indeed, in our current study, the spatially uniform MFM could generate high and low efficiency states in the training set, but not the test set (figure below). On the other hand, the

gradient-informed MFM was able to capture the high and low efficiency states in the test set.

We have updated the paper as follows:

Figure S12. High and low efficiency states in time-varying FC. (A) Empirical FCD from a participant from the HCP test set. The high and low coherent states in the current study corresponded to periods of high and low efficiency first discovered by Zalesky and colleagues (Zalesky et al., 2014). (B) Spatially heterogeneous pMFM was able to generate both high and low efficiency states in the test set. (C) Spatially homogeneous pMFM was able to generate high and low efficiency states in the

training set. (D) Spatially homogeneous pMFM was not able to generate high and low efficiency states in the test set.

pg 25 (Results)

Fourth, although time-varying FC was represented using the FCD matrix (Figure 2A), other representations could be possible. Zalesky and colleagues explored time-varying FC by computing time-varying network efficiency for each sliding window (Zalesky et al., 2014). They found high and low efficiency states, which appeared to correspond to the high and low coherent states in the FCD matrix (Figure S12A). The pMFM also captured these high and low efficiency states in test set (Figure S12B). On the other hand, the spatially homogeneous MFM could generate high and low efficiency states in the training set, but not the test set (Figures S12C & S12D).

(RIQ8) More substantially, deeper insights into the dynamics could be obtained by looking at the relationship of the amplitude and variance in these two 'modes', and also from examining the lifetime distributions – both as per Freyer et al (2012), already cited. Ideally this would encompass the entire distribution of states (/correlations) and can be used to disambiguate meta- from multi-stability. As it stands, partitioning the data into top and bottom deciles and performing statistics on these seems quite circular.

We thank the reviewer for the suggestion. We have now performed a similar analysis (Freyer et al., 2012) in which we fitted a mixture of two Gaussian distributions to delimit the two states. We find that the dwell time distributions are similar between the simulated and empirical data and appear to follow an exponential distribution. We have updated the manuscript as follows.

pg. 18-19 (Results)

To quantify this phenomenon, for each run of each participant in the HCP test set, we fitted a mixture of two Gaussian distributions to the histogram of the FCD mean (Freyer et al., 2012). The cross-over point of the two Gaussian distributions was used to threshold the FCD mean. Time points with FCD mean greater than the threshold were designated as the coherent state (high FCD mean), while time points with FCD mean lower than the threshold were designated as the incoherent state (low FCD mean). The SW-STD was then averaged across all cortical regions and all runs of each participant. As shown in Figure 5D, the SW-STD was significantly higher during the coherent state than the incoherent state ($p = 6.9e-150$). Similar results were obtained for the pMFM simulations (Figure 5E). The dwell time distributions of the two states were also similar between the empirical and simulated data (Figure S7). The two distributions appeared to follow an exponential distribution (as opposed to a Gamma distribution), suggesting the presence of multi-stability rather than meta-stability (Cocchi et al., 2017).

Figure S7. Distributions of dwell time for coherent (high FCD mean) and incoherent (low FCD mean) states in both empirical and pMFM-simulated results. The dwell time distributions of the two states were similar between empirical and simulated data. Furthermore, the two distributions appeared to follow an exponential distribution (as opposed to a gamma distribution), suggesting the presence of multi-stability rather than meta-stability (Cocchi et al., 2017).

(RIQ9) Lines 374-5 – these should all refer to Figure 6, not 7

Thank you for pointing out the typos. We have fixed them.

pg 20 (Results)

The neural signals of the top five FCD-STD regions (sensory-motor drivers; Figure 6B) were then perturbed to increase their amplitude. The perturbation led to the successful transition of the FCD into a more coherent state with higher FCD mean ($p = 6e-14$; Figure 6D). Perturbation of the bottom five FCD-STD regions (Figure 6B) did not lead to an increase in FCD mean.

Reviewer #2:

(R2Q1) This study simulates static and dynamic functional connectivity using group-level information about functional gradient and myelination to inform heterogeneity across the cortex. The simulated results show improved correspondence to empirical data compared with a model that assumed homogeneity. The findings furthermore suggest the presence of high and low coherence states, which were causally linked to signal amplitude in sensory-motor regions in follow-up analyses. Associations between spatial amplitude patterns and gene expression were also shown..

This is a well-written and very thorough study that provides important new insights into neural and genetic mechanisms that shape patterns of functional connectivity. The control analyses are appropriate and preemptively addressed most of the questions that came to my mind while reading the article.

We thank the reviewer for the positive comments.

(R2Q2) The functional gradient maps essentially reflect the principal direction of spatial organization in static functional connectivity. These maps are not independent from the static FC estimated in this paper and there is a potential concern of circularity in their use to simulate static FC. This is consistent with the results in Fig 3C showing good performance in predicting static FC using the gradient maps alone. This is not a major issue and it is actually of interest to bridge these different representations of static FC (i.e., gradients vs parcellated FC). Nevertheless, this should be clarified and discussed in the manuscript to help the reader in interpreting the results.

We thank the reviewer for this comment. We agree that the FC gradient appeared to be better than the T1w/T2w map at explaining static FC. This makes sense given that the FC gradient was derived from static FC. However, we note that the results were not circular given that the FC gradient was derived from the training set and performance was evaluated in the test set.

To elaborate more, this situation is entirely analogous to the use of resting-state parcellations for resting-state FC analysis. For example, the Schaefer parcellation was derived from the resting-state FC of 744 participants. In one analysis (Schaefer et al., 2018), we showed that vertices within Schaefer parcels exhibited more similar resting-state fMRI time courses than other parcellations in a second set of 745 participants (Schaefer et al., 2018). There was no circularity in this analysis because the parcellations were estimated and tested on two non-overlapping groups of participants. Similarly, in the current study, the MFM parameters were estimated and then tested on non-overlapping groups of participants. Therefore, while it makes intuitive sense that parameterizing the MFM using resting-FC would help to explain out-of-sample resting-FC better, it is not circular.

We have updated the manuscript as follows.

pg. 8 (Results)

Both FC gradient and T1w/T2w map were estimated from the training set.

pg 13 (Results)

On the other hand, FC gradient was better than T1w/T2w map at explaining static FC across the three experiments (Figures 3 and S4), which made intuitive sense given that FC gradient was derived from static FC. However, we note that the analyses were not circular given that the FC gradient was derived from the training set and performance was evaluated on the test set.

pg. 29 (Discussion)

Furthermore, while it made intuitive sense that utilizing the resting-state FC gradient would help to explain resting-fMRI dynamics, the training-validation-test scheme ensured the analysis was not circular.

(R2Q3) Weaker agreement is shown between the proposed model and several other tested models (assumed spatial homogeneity/ gradient only/ T1-T2 only etc; Figs 2&3). However, no statistical testing is performed for some of these comparisons. I would encourage the authors to include statistical comparisons.

We thank the reviewer for the suggestion. Summary statistics and p values are now reported in Table S1. The pMFM was statistically better than the spatially homogeneous MFM for both metrics (FC correlation and FCD KS statistic) in the test set. Compared with other alternative parameterizations, pMFM was statistically better in one metric and statistically comparable in the other metric. We have updated the manuscript as follows.

pg 11 (Results)

Table S1 provides summary statistics of the two metrics (FC correlation and FCD KS statistic) in the training, validation, and test sets. The pMFM was statistically better than the spatially homogeneous MFM for both metrics in the test set. Compared with other alternative parameterizations, pMFM was statistically better in one metric and statistically comparable in the other metric.

	FC Correlation Training	FC Correlation Validation	FC Correlation Test	FC p-value Test Set	FCD KS Statistics Training	FCD KS Statistics Validation	FCD KS Statistics Test	FCD p- value Test
pMFM	0.66±0.03	0.67±0.03	0.66±0.03		0.04±0.02	0.03±0.01	0.12±0.03	
Gradient only	0.63±0.06	0.63±0.06	0.63±0.06	0.188	0.15±0.15	0.20±0.18	0.31±0.26	0.031
T1w/T2w only	0.56±0.04	0.56±0.04	0.55±0.03	2.0e-5	0.07±0.05	0.09±0.05	0.15±0.06	0.123
Constant w, I & σ	0.58±0.03	0.55±0.04	0.55±0.05	2.4e-5	0.28±0.30	0.31±0.29	0.50±0.31	0.001
Constant w	0.63±0.03	0.64±0.03	0.62±0.03	0.0245	0.05±0.02	0.07±0.04	0.11±0.04	0.729
Constant I	0.58±0.06	0.59±0.07	0.58±0.06	0.0037	0.11±0.06	0.13±0.08	0.14±0.06	0.179
Constant σ	0.57±0.03	0.59±0.04	0.57±0.04	2.48e-5	0.09±0.03	0.08±0.03	0.24±0.21	0.088
Non- parametric	0.66±0.02	0.66±0.03	0.66±0.03	0.754	0.06±0.03	0.04±0.02	0.28±0.11	2.1e-4
FC Cost only	0.74±0.005	0.76±0.002	0.73±0.01	2.1e-6	0.87±0.005	0.87±0.005	0.88±0.01	7.1e-23

Table S1. Model performance in the HCP training, validation, and test sets. The HCP training set was used to generate 5000 MFM parameter sets, which were then evaluated in the HCP validation set. The top 10 candidates from the validation set were then evaluated in the HCP test set. The table below shows the performance (mean \pm std) of these top 10 candidates in the HCP training, validation and test sets. Higher FC correlations and lower FCD KS statistics indicate better fit. The last column showed the p values of two-sample two-tail t-tests between pMFM and other parameterizations or cost function in the test set. Bolded results indicate statistically significant results after false discovery rate (FDR) correction ($q < 0.05$). Our approach (pMFM) was statistically better than (or comparable to) all other parameterizations in terms of FC correlation and/or FCD KS statistic in the test set. Optimizing for only fit to static FC (last row in table) led to slightly better (but statistically significant) fit to static FC in the test set, but much worse (and statistically significant) fit to FCD in the test set.

(R2Q4) Group-level gradient and T1/T2 maps are used to inform the model. This is appropriate for this first study, which is already comprehensive in many other respects. But I wonder if the use of group-level maps limits the degree to which simulated data are able to capture between-subject variability present in empirical data. This is an interesting area for future research and I would be interested in the author's thoughts on this front. Perhaps it is something to consider including in the discussion.

We thank the reviewer for this suggestion. We are actively working on individual-level pMFM as a separate study. But given the reviewer's request, we have added some preliminary results of individual-level pMFM. We found that even at the

individual-level, combining T1w/T2w map and FC gradient yielded more realistic static FC and FCD than using T1w/T2w map or FC gradient alone.

However, we note that this analysis still utilized group-level FC gradient, T1w/T2w map and SC matrices, although the model was fitted and evaluated using the data of individual participants. So far, we have not found that individual-level FC gradient, T1w/T2w map and/or SC matrix led to greater benefits (not shown). However, given that the research is quite preliminary, we decided not to mention the lack of benefits in the manuscript.

pg 26 (Results)

Finally, to explore the possibility of individual-level pMFMs, we considered 12 participants from the HCP test-retest dataset that overlapped with our test set. There were four MRI sessions for each participant. The first two sessions and the last two sessions were on average 3.8 ± 1.5 months apart. Similar to previous analyses, the pMFM was optimized using group-level FC gradient and group-level T1w/T2w map from the training set. The main difference is that the model was optimized using group-level SC from the test set, as well as static FC and FCD from the first two sessions of individual participants. The top 10 parameter sets from the first two sessions were then evaluated in the remaining two sessions. We found that combining T1w/T2w map and FC gradient yielded more realistic static FC and FCD than using T1w/T2w map or FC gradient alone at the individual level. Future studies will explore whether individual-level FC gradient, T1w/T2w map and SC could bring further benefits to individual-level MFMs.

Figure 8. At the individual-level, pMFM parameterized by both group-level FC gradient and T1w/T2w map yielded more realistic static FC and FCD than FC gradient or T1w/T2w map alone. (A) Agreement (Pearson's correlation) between simulated and empirically observed static FC in the test sessions of individual participants. (B) Disagreement (KS distance) between simulated and empirically observed FCD in the test sessions of individual participants. (C) Total cost in the test sessions of individual participants. * indicates statistical significance after correcting for multiple comparisons with a false discovery rate of $q < 0.05$.

(R2Q5) The first functional gradient is used to inform spatial heterogeneity. I would be interested in how using additional gradients may influence the results, especially in relation to the amplitude-state associations. Again, I appreciate that this might be outside of the scope of the current paper and more a direction for future research.

We thank the reviewer for the suggestion. We have now added a control analysis that considered alternate gradient maps, including the second FC gradient (Margulies et al., 2016), inter-subject functional connectivity variability map (Mueller et al., 2013), first structural covariance gradient (Valk et al., 2020) and the first genetic principal component.

Interesting, the first principal gradient alone did not lead to the best performance. The best single parameterization was the T1w/T2w map. Combining T1w/T2w map with the first FC gradient (i.e., original pMFM) led to the best performance in the test set, but the improvement was not significant when the T1w/T2w map was replaced with inter-subject FC variability, first genetic principal component or second FC gradient. We found that in these cases, the resulting amplitude-state associations (i.e., FCD-STD correlation maps) remained highly similar to the original FCD-STD map with correlations greater than 0.9.

Changes to the manuscript can be found in our response to R3Q2.

Reviewer #3:

(R3Q1) This manuscript describes a new computational modeling approach which seeks to estimate observed patterns of static and dynamic cortical functional connectivity (FC and FCD respectively) from group-level estimates of structural connectivity, and region-specific cortical myelination (T1w/T2w ratio) and a primary FC spatial gradient. The major advance of this model on prior work is that it models observed FC and FCD using a spatially heterogeneous cortical mean field model where key local circuit properties are parameterized using the local group-average measures of myelination and primary FC gradient. The authors show that use of region-specific information improved model fit. They then use the winning models to test hypotheses re the role of amplitude variability in rsfMRI signals in “flipping” between periods of high and low-coherence in FCD - concluding that sensorimotor regions may drive this behavior. Finally, the authors show that regional variation in rsfMRI signal amplitude from their computational model is correlated with regional variation in the difference in molecular markers of parvalbumin and somatostatin interneurons from the Allen Human Brain Atlas.

I think this study will be of interest to the research community, and is notable for several strengths: the authors pursue models that are more realistic than prior ones, by virtue of allowing local circuit features to vary across the cortical sheet; there is a rigorous approach to model selection with extensive sensitivity analyses to test the influence of important methodological decisions; the linkage to gene-expression data is also of value, and the authors do this with a sensible discussion of caveats and limitations. The authors of this work are leaders in the generation and implementation of biophysical models for the brain, so I would anticipate that this aspect of the work is sound. However, I do not have expertise in biophysical modeling of brain activity, and it would be important that this expertise is represented within the peer-review process

We thank the reviewer for positive comments.

(R3Q2) The T1w/T2w and FC gradient are just two of many cortical features that all capture the same “primary gradient” spanning sensorimotor and cortical. Others would be structural/functional variability across people, first PC of expression or sets of genes loading highly onto it etc. Is there a way of running “control” versions of the model that swap out T1w/T2w and/or FC gradient for these alternatives - to show that these two features perform best. These two features are sensible to consider given the model features of recurrent connections, external input and noise. But the set-up in this paper provides a nice opportunity to pit facets of the primary gradient against each other.

We thank the reviewer for the suggestion. We have now added a control analysis that considered alternate gradient maps, including the second FC gradient (Margulies et

al., 2016), inter-subject functional connectivity variability map (Mueller et al., 2013), first structural covariance gradient (Valk et al., 2020) and the first genetic principal component.

Interesting, the first principal gradient alone did not lead to the best performance. The best single parameterization was the T1w/T2w map. Combining T1w/T2w map with the first FC gradient (i.e., original pMFM) led to the best performance in the test set, but the improvement was not significant when the T1w/T2w map was replaced with inter-subject FC variability, first genetic principal component or second FC gradient.

We have updated the manuscript as follows.

pg 24 (Results)

We also repeated the training-validation-test procedure with alternate gradient maps, including the second FC gradient (Margulies et al., 2016), inter-subject functional connectivity variability map (Mueller et al., 2013), first structural covariance gradient (Valk et al., 2020) and the first genetic principal component (Figure S10). To provide additional context, Figure S10 shows the correlations among the different gradient maps and the top estimated model parameters (w , I , σ) from the original pMFM (Figure 4).

The estimated model parameters were most strongly correlated with the first principal gradient, although we note that the first principal gradient alone did not lead to the best performance in the test set (Figure S10). Instead, the best single parameterization was the T1w/T2w map. Combining T1w/T2w map with the first FC gradient (i.e., original pMFM) led to the best performance in the test set, but the improvement was not significant when the T1w/T2w map was replaced with inter-subject FC variability, first genetic principal component or second FC gradient. However, we note that in these cases, the resulting FCD-STD correlation maps remained highly similar to the original FCD-STD map (Figure 6B) with correlations greater than 0.9, suggesting that these cortical features may index similar underlying mechanisms.

Figure S10. In addition to T1w/T2w map and first FC gradient, we considered other alternative gradient maps, including second FC gradient (Margulies et al., 2016), inter-subject functional connectivity variability map (Mueller et al., 2013), first structural covariance gradient (Valk et al., 2020) and the first genetic principal component. All parameterizations were optimized using the same training-validation-test procedure in Figure 1B. * indicates that parametrization with T1w/T2w map and first FC gradient was statistically better after multiple comparisons correction with a false discovery rate of $q < 0.05$. In general, inclusion of the first FC gradient was important for performance in the test set. Although combining T1w/T2w map with the first FC gradient led to numerically the best performance in the test set, the improvement was not significant when the T1w/T2w map was replaced with inter-subject FC variability, first genetic principal component or second FC gradient.

Figure S11. Correlations among the top estimated model parameters (w , I , σ) from the original pMFM (Figure 4), first FC gradient, first structural covariance gradient (Valk et al., 2020), T1w/T2w map, first genetic principal component, PVALB-SST, inter-subject functional connectivity variability map (Mueller et al., 2013), and second FC gradient (Margulies et al., 2016). The first FC gradient was strongly correlated with the estimated model parameters (w , I , σ), while the T1w/T2w map was strongly correlated with the first genetic principal component consistent with previous studies (Burt et al., 2018).

(R3Q3) The T1w/T2w and FC gradient seem to be making differential contributions for static FC (where FC does most of the "work") vs. FCD (where T1/T2 seems to get you pretty close to the combined model). Would be good to see some more analytic exploration and/or Discussion of this.

The reviewer has raised a very good point. Figure 3 does seem to suggest that the FC gradient and T1w/T2w map contribute to static FC and FCD respectively. However, we note that our cost function weighed FC and FCD equally: $(1 - r) + KS$. When the relative weights of FC and FCD were altered, combining both T1w/T2w map and FC gradient still yielded better test set performance than either T1w/T2w map or FC gradient alone (see new Figure S4 below). However, T1w/T2w map alone was no longer better than FC gradient at explaining FCD. On the other hand, FC gradient was better than T1w/T2w map at explaining static FC across all experiments. We have updated the manuscript as follows.

pg. 13 (Results)

In all the previous analyses, the overall cost was defined as $(1 - r) + KS$, which placed equal weights on fitting FC and FCD. When the relative weights of FC

and FCD were altered, combining both T1w/T2w map and FC gradient still yielded better test set performance than either T1w/T2w map or FC gradient alone (Figure S4). Although the original analysis (Figure 3) suggested that T1w/T2w map explained FCD better than FC gradient, this was no longer the case when the relative weights were altered (Figure S4). On the other hand, FC gradient was better than T1w/T2w map at explaining static FC across the three experiments (Figures 3 and S4), which made intuitive sense given that FC gradient was derived from static FC. However, we note that the analyses were not circular given that the FC gradient was derived from the training set and performance was evaluated on the test set.

Figure S4. Combining both T1w/T2w map and FC gradient yielded better performance even when the relative weights of static FC cost and FCD cost in the cost

function were altered. (A) Stronger weight on FCD cost. (B) Stronger weight on FC cost. Not surprisingly, increasing the weight of the FC cost (panel B) yielded better (greater) static FC correlation and worse (greater) FCD KS statistic in the test set. * indicates statistical significance after multiple comparisons correction with false discovery rate (FDR) $q < 0.05$.

(R3Q4) I correlation matrix for inter-regional variation in T1w/T2w, FC gradient, w , I and σ would be good to include (as sup maybe) -perhaps also alongside some of the other imaging and expression-derived features mentioned in comment #1 above. My general point is that many of these results seem to hit on the same primary gradient, and I think this means we really need to see just how different the input maps, parameter maps (Fig 4) and others (slide 6, 7) are.

We thank the reviewer for the suggestion. We have now included the correlation matrix. Please see our response to R3Q2.

(R3Q5) Inter-individual variation: This may reflect a limitation in my understanding of the method, but ... it seemed to me that the first section on predicting FC and FCD from structural connectivity, T1w/T2w and FC gradient through the dynamic mean field model could be done at the individual level too? If this is correct, then that would provide a good way to test the stability of each feature's utility to the model across people, and provide an added level of association.

The reviewer is correct that the modeling could potentially be performed at the individual level. We are actively pursuing this research as a separate study. Given the reviewer's request, we have added some preliminary results. We found that even at the individual-level, combining T1w/T2w map and FC gradient yielded more realistic static FC and FCD than using T1w/T2w map or FC gradient alone.

However, we note that this analysis still utilized group-level FC gradient, T1w/T2w map and SC matrices, but the model was fitted and evaluated using the data of individual participants. So far, we have not found that individual-level FC gradient, T1w/T2w map and SC matrix led to greater benefits (not shown). However, given that the research is quite preliminary, we decided not to mention the lack of benefits in the manuscript.

Changes to the manuscript can be found in our response to R2Q4.

REVIEWERS' COMMENTS

Reviewer #1 (Remarks to the Author):

The authors have addressed my prior concerns with excellent revisions, new work and a clear rebuttal. This is an exciting paper that further advances the nascent field of mean field modelling of the brain toward maturity. I recommend the paper for publication with only a minor text correction that could be done in the proofing stage

I would suggest to delete the phrase "first discovered by Zalesky and colleagues (Zalesky et al., 2014)" - precedence claims are (I'm sure the authors would agree) somewhat meaningless. The point in my last remark - and which the authors have known nicely shown - is convergence of findings across studies/techniques to help integrate the field, rather than who first described something.

Reviewer #2 (Remarks to the Author):

The authors adequately addressed my comments. I particularly like the new control analyses whether the authors test the second gradient, inter-subject variability, structural covariance gradient, and genetic principal component.

Reviewer #3 (Remarks to the Author):

The authors have fully responded to all my comments on initial review and I would support publication of this paper in its current form

Response to reviewers NCOMMS-21-07715 “Anatomical and Functional Gradients Shape Dynamic Functional Connectivity in the Human Brain”

Reviewer #1:

(R1Q1) The authors have addressed my prior concerns with excellent revisions, new work and a clear rebuttal. This is an exciting paper that further advances the nascent field of mean field modelling of the brain toward maturity. I recommend the paper for publication with only a minor text correction that could be done in the proofing stage.

I would suggest to delete the phrase "first discovered by Zalesky and colleagues (Zalesky et al., 2014)" - precedence claims are (I'm sure the authors would agree) somewhat meaningless. The point in my last remark - and which the authors have known nicely shown - is convergence of findings across studies/techniques to help integrate the field, rather than who first described something.

We thank the reviewer for the positive comments. We have removed the phrase from Figure S12 as requested.

Reviewer #2:

(R2Q1) The authors adequately addressed my comments. I particularly like the new control analyses whether the authors test the second gradient, inter-subject variability, structural covariance gradient, and genetic principal component.

We thank the reviewer for the positive comments.

Reviewer #3:

(R3Q1) The authors have fully responded to all my comments on initial review and I would support publication of this paper in its current form

We thank the reviewer for the positive comments.